# Interaction between Microalgae *P. tricornutum* and Bacteria *Thalassospira* sp. for Removal of Bisphenols from Conditioned Media

**DOI:** 10.3390/ijms23158447

**Published:** 2022-07-30

**Authors:** David Škufca, Darja Božič, Matej Hočevar, Marko Jeran, Apolonija Bedina Zavec, Matic Kisovec, Marjetka Podobnik, Tadeja Matos, Rok Tomazin, Aleš Iglič, Tjaša Griessler Bulc, Ester Heath, Veronika Kralj-Iglič

**Affiliations:** 1University of Ljubljana, Faculty of Health Sciences, Biomedical Research Group, Zdravstvena 5, SI-1000 Ljubljana, Slovenia; david.skufca@bia.si (D.Š.); daria.bozic@gmail.com (D.B.); marko.jeran@ijs.si (M.J.); tjasa.bulc@zf.uni-lj.si (T.G.B.); 2University of Ljubljana, Faculty of Electrical Engineering, Laboratory of Physics, Tržaška 25, SI-1000 Ljubljana, Slovenia; ales.iglic@fe.uni-lj.si; 3Department of Physics and Chemistry of Materials, Institute of Metals and Technology, Lepi Pot 11, SI-1000 Ljubljana, Slovenia; matej.hocevar@imt.si; 4Department of Molecular Biology and Nanobiotechnology, National Institute of Chemistry, Hajdrihova 19, SI-1000 Ljubljana, Slovenia; polona.bedina@ki.si (A.B.Z.); matic.kisovec@ki.si (M.K.); marjetka.podobnik@ki.si (M.P.); 5University of Ljubljana, Faculty of Medicine, Institute of Microbiology and Immunology, Zaloška 4, SI-1000 Ljubljana, Slovenia; tadeja.matos@mf.uni-lj.si (T.M.); rok.tomazin@mf.uni-lj.si (R.T.); 6Department of Environmental Sciences, Jožef Stefan Institute, Jamova 39, SI-1000 Ljubljana, Slovenia; ester.heath@ijs.si; 7Jožef Stefan International Postgraduate School, Jamova 39, SI-1000 Ljubljana, Slovenia

**Keywords:** contaminants of emerging concern, bisphenol removal, microalgae, *Phaeodactylum tricornutum*, bacteria, extracellular vesicles, small cellular particles, electron microscopy of small cellular particles, nanoalgosomes

## Abstract

We studied the efficiency of three culture series of the microalgae *Phaeodactylum tricornutum* (*P. tricornutum*) and bacteria *Thalassospira* sp. (axenic microalgae, bacterial culture and co-culture of the two) in removing bisphenols (BPs) from their growth medium. Bacteria were identified by 16S ribosomal RNA polymerase chain reaction (16S rRNA PCR). The microorganism growth rate was determined by flow cytometry. Cultures and isolates of their small cellular particles (SCPs) were imaged by scanning electron microscopy (SEM) and cryogenic transmission electron microscopy (Cryo-TEM). BPs were analyzed by gas chromatography coupled with tandem mass spectrometry (GC-MS/MS). Our results indicate that some organisms may have the ability to remove a specific pollutant with high efficiency. *P. tricornutum* in axenic culture and in mixed culture removed almost all (more than 99%) of BPC2. Notable differences in the removal of 8 out of 18 BPs between the axenic, mixed and bacterial cultures were found. The overall removals of BPs in axenic *P. tricornutum*, mixed and bacterial cultures were 11%, 18% and 10%, respectively. Finding the respective organisms and creating microbe societies seems to be key for the improvement of wastewater treatment. As a possible mediating factor, numerous small cellular particles from all three cultures were detected by electron microscopy. Further research on the mechanisms of interspecies communication is needed to advance the understanding of microbial communities at the nano-level.

## 1. Introduction

Nature-based solutions for wastewater treatment with their ability for integrated resource management are promising for developing a circular economy in the urban environment [1]. One such example is algal photobioreactors, most notably, high-rate algal ponds (HRAP), which rely on algae and bacterial communities to treat wastewater and produce biomass [2]. Compared to activated sludge reactors used for treating wastewater, HRAPs have longer hydraulic retention times and a large surface area, i.e., 1 ha or more in full-size HRAP systems [2,3,4]. In addition, they do not require active aeration since the algae produce O_2_ and organic acids needed by the bacteria, which contribute CO_2_ and nutrients for the algae [2]. The main advantage of photobioreactors is the production of nutrient- and energy-rich algal biomass that may be exploited as a feedstock, i.e., for polymer, energy (e.g., biogas or biodiesel) and fertilizer production [5,6,7,8]. However, our understanding of nature-based solutions needs to progress from the technological unit level to the cellular community level since cellular communication plays a fundamental role in the homeostasis of complex biological systems where synchronization, cooperation, quick adaptation and specialization/differentiation of the cells occurs [9,10,11]. It is now acknowledged that cells release various types of SCPs, including extracellular vesicles (EVs), antibody complexes, lipoproteins and other particles capable of transporting different substances, such as proteins, lipids, sugars and nucleic acids [12]. EVs have been implicated in many aspects of cell physiology, such as stress response, intercellular competition, lateral gene transfer (via RNA or DNA), pathogenicity and detoxification [12]. Although microalgal SCPs were first observed in the 1970s [13,14], and have recently become a subject of further interest [15,16,17], their roles in communities are not yet fully understood.

Contaminants of emerging concern (CECs) include active components of human and veterinary pharmaceuticals, illicit drugs, personal care products, pesticides, hormones, flame retardants, plasticizers and other compounds, as well as their metabolites and transformation products (TPs) [18,19]. However, their environmental occurrence and fate have been investigated only recently due to awareness of potential adverse ecological and human health impacts, although CECs may not be new in the environment [20]. CECs are typically present in the environment at trace levels, and only recent advances in analytical instrumentation have allowed their detection at low concentrations (ng/L and even pg/L) [18]. Bisphenols (BPs) are a group of CECs characterized by two hydroxyphenyl groups bound by a hydrocarbon bridge and otherwise containing diverse chemical groups, resulting in different physicochemical properties and consequent environmental behaviour, making them suitable model compounds. BPs are monomers used to produce polycarbonate, epoxy resin, polysulfone, polyacrylate, polyetherimide, and as an additive in thermal paper, polyvinyl chloride and other products [21], and it was indicated that their emissions into wastewater are not negligible [22]. Studies point toward BPs causing endocrine disruption and other toxic effects (e.g., reproductive toxicity), neurotoxicity and cytotoxicity [23], which is concerning as they may cause ecological harm [24]. Wastewater represents the main influx of CECs to the environment due to inadequate removal during wastewater treatment [2,19,25]. They may also pose a risk to humans when considering reusing treated wastewater products (e.g., reclaimed water and biomass) for activities such as agriculture.

Microalgal photobioreactors are an alternative to conventional wastewater treatment [26]. Biodegradation of CECs in microalgal photobioreactors results from the metabolism of microalgae and bacteria, either intracellularly or extracellularly [27]. Co-metabolic biodegradation may be accomplished by non-specific enzymes produced to assimilate other organic compounds [28]. Furthermore, microalgae often grow in co-culture with bacteria [29]. Liu et al. (2021) postulated that biodegradation of CECs may take place according to three scenarios: (1) microalgae do not directly degrade the compound but provide a favourable environment for bacteria, promoting biodegradation, (2) bacteria and microalgae both significantly and directly contribute to the biodegradation of the CEC and (3) microalgae and bacteria synergistically degrade the CEC, where one can degrade the intermediate products of the other or vice versa [28]. Past experimental studies have shown that a co-culture of microalgae and bacteria is more efficient at removing organic pollutants than a single culture. Similarly, Ji et al. (2018) showed that a co-culture of *Chlorella vulgaris* (*C. vulgaris*) and *Bacillus licheniformis* reduces chemical oxygen demand (COD), total dissolved nitrogen and total dissolved phosphorus compared to an axenic culture of microalgae; additionally, they reported a two-fold higher peak in chlorophyll *a* values in the co-culture, along with altered expressions of chlorophyll-related genes [30].

Some bacterial strains in the native phycosphere (mini ecosystem-surrounding microalgal cell walls) may improve the growth of *C. vulgaris* (e.g., *Flavobacterium*, *Hyphomonas*, *Rhizobium* and *Sphingomonas*). In contrast, others may be inhibitory (*Microbacterium* and *Exophiala*), illustrating that not all interactions need to be mutualistic. Kumari et al. (2016) reported that a co-culture of *Scenedesmus* sp. and *Paenibacillus* sp. was more successful than axenic culture in removing organic contaminants, total dissolved solids (TDS), COD and heavy metals, as well as showing the highest reduction in cytotoxicity and genotoxicity [31]. These studies point to interspecies interactions between bacteria and microalgae, which may impact a photobioreactor’s performance, although the underlying mechanisms are poorly understood. It is therefore of utmost importance to enhance our understanding of intercellular communication mechanisms, and we expect that this would have implications across multiple fields of science [17].

In this work, we attempted to elucidate the mechanisms underlaying the removal of BPs by microalgae. Based on previous results [26,32], we hypothesized that a co-culture of microalgae and bacteria would be more efficient in removing BPs than a pure axenic microalgal or bacterial culture, due to greater expected biomass and potential mutualism in a mixed culture. Three different in vitro cultures were studied: (1) an axenic microalgal culture of *P. tricornutum*, (2) a co-culture of microalgae and bacteria and (3) a bacterial culture. We wished to see the ability of microalgae to remove pollutants in the presence of bacteria and without bacteria, and seek possible connections between these activities. As a possible underlying mechanism, we considered the mediated interaction between the microorganisms by SCPs. To fulfil our goals, we followed the abundance of microorganisms in the culture during growth and determined the concentrations of BPs in the conditioned media. We isolated SCPs from the conditioned media and observed them by SEM and Cryo-TEM. The study aims to connect the role of SCPs as mediators of intercellular interactions to the broader field of ecology. To our best knowledge, in this work the above issues were addressed for the first time.

## 2. Results

### 2.1. Culture Growth

Cultures are referred to as experimental series (addition of BPs): axenic microalgae (EA), co-culture of microalgae and bacteria (EC), bacterial culture (EB) and blank control series (no addition of BPs): axenic microalgae (BA), co-culture of microalgae and bacteria (BC) and bacterial culture (BB), each in triplicate (for details see Section 4.1 and Table 1). The growth of microalgal cells in different cultures was followed by counting events of autofluorescent particles (AFPs) by flow cytometry (FCM), which also sets them apart from non-fluorescent particles (NFPs), which were attributed to bacteria, as well as to SCPs from bacteria and microalgae. Both AFPs and NFPs were quantified in all experimental and blank control series to check for possible contamination by microalgae or bacteria. The AFPs and NFPs were defined by the respective regions of the FCM diagrams as described in Section 4.2.

Quantification of AFPs shows that the starting concentrations of microalgal cells were approximately 10^6^ cells mL^−1^ for the microalgae (Figure 1 EA, BA, EC, BC), whereas the concentration of AFPs in EA showed a microalgal cell concentration of (mean ± standard deviation) 5.3 ± 2.8 × 10^6^ cells mL^−1^ and a similar value of 5.0 ± 2.2 × 10^6^ cells mL^−1^ in BA after 168 h (Figure 1). EC and BC reached higher microalgal cell concentrations than EA and BA, i.e., 7.6 ± 1.4 × 10^6^ cells mL^−1^ and 7.3 ± 1.7 × 10^6^ cells mL^−1^, respectively.

The starting concentration of detected NFPs in the axenic cultures was 2.2 × 10^5^ mL^−1^ in EA and 2.3 × 10^5^ mL^−1^ in BA. NFP concentration increased during the experiment for less than 5.0 × 10^5^ mL^−1^, an increase attributed to SCPs and cellular debris. The starting concentration of events detected in the same gate in the co-cultures and bacterial cultures was approximately 0.5 × 10^6^ mL^−1^. Therefore, events detected in this gate in the samples with bacteria present were mainly attributed to the bacterial cells.

After 168 h, the concentration of NFPs was comparable in the BP-treated and non-treated replicates: 1.9 ± 0.2 × 10^6^ mL^−1^ in EC, 2.0 ± 0.1 × 10^6^ cells mL^−1^ in BC, 8.0 ± 0.6 × 10^6^ cells mL^−1^ in EB and 10.3 ± 2.0 × 10^6^ cells mL^−1^ in BB. In these cultures, the concentration of NFPs increased up to 72 h and then declined slightly until 168 h. The maximum concentrations (measured at 72 h) were 3.6 ± 0.7 × 10^6^ mL^−1^ and 2.8 ± 0.2 × 10^6^ mL^−1^, in EC and BC, respectively, and 10.3 ± 0.4 × 10^6^ mL^−1^ and 10.6 ± 1.1 × 10^6^ mL^−1^, in EB and BB, respectively (Figure 1).

### 2.2. Cell Morphology and SCP Characterization

In microalgae-containing samples, *P. tricornuum* was mainly in the fusiform shape (Figure 2 EA, BA, EC, BC; white arrows). Bacteria were observed in co-culture and bacterial samples (Figure 2 EC, BC, EB, BB; dashed white arrows). Bacterial cells with distinct morphologies were observed, such as rod-shaped cells (Figure 3 BB; dashed white arrow), rod-shaped cells with a rough hairy surface (Figure 3 EB; fat white arrow) and drill-shaped cells (Figure 3 EC; white arrow). The bacteria were identified as *Thalassospira* sp. The members of the genus *Thalassospira* are Gram-negative rod-shaped bacteria from the *Alphaproteobacteria*. We observed no effects of BPs on the morphology in any of the three types of cultures (microalgal, bacterial or co-culture, Figure 2).

Isolates of SCPs from the microalgal- and co-culture-conditioned media were rich in globular particles with a distinctly rough surface (Figure 4E and Appendix A). Such structures were observed in some regions of microalgal cells with a rough, nanostructured epitheca (e.g., Figure 2 EA) in all cultures containing microalgae (EA, BA, EC and BC). Cryo-TEM microscopy of an isolate from an axenic microalgal culture showed two types of SCPs: electron-dense clusters and membrane-enclosed EVs (Figure 4F, black and white arrowheads, respectively). The membrane of EVs was surrounded by radially oriented fibres, forming an approximately 20 nm-thick coat, visible in Cryo-TEM images (Figure 4F). In the isolates from bacterial cultures, we observed singular SCPs and buds having a smooth globular shape (Figure 4A–D).

### 2.3. Bisphenol Residue Mass Balance and Removal

Concentrations of BPs in either the abiotic light (ABL) or abiotic dark (ABD) controls showed no appreciable decrease between 72 h and 168 h (Appendix A). Likewise, no significant differences were observed between the ABL and ABD series, indicating that no photodegradation took place (Appendix A). Their higher hydrophobicity can also explain the lower concentrations of BPFL–BPPH than their nominal starting concentrations (1000 ngL^−1^) as a result of adsorption (49%) to the glass walls of the vessel [32]. The blank controls (BA, BC and BB) indicated no contamination with BPs, except for BPA, which reached 100 ngL^−1^ in one BA microalgae culture (not shown). However, determining BPA accurately at low levels can be problematic due to background contamination [33].

After 168 h, the proportion of BPs residues in biomass was highest for the more hydrophobic compounds (log P 5.99–7.34) BPFL, BPBP, BPM, BPP and BPPH, and ranged from 10 ± 4% (BPFL in axenic microalgae) to 30 ± 4% (BPPH in bacteria, Figure 5). Other more polar BPs (BPS–BPAP, log P 2.32–5.18) were predominantly found in the aqueous phase (>60%, Figure 5). The proportion of BPs in the aqueous phase ranged from <limit of quantitation (LOQ) (BPC2 in axenic microalgae and co-culture) to 81 ± 5% (BPAF in axenic microalgae). Total removal ranged from 17 ± 5% (BPAF in axenic microalgae) to >99% (BPC2 in axenic microalgae and co-culture). Removals between the cultures most notably differed for BPC2, BPC and BP26DM.

Biotic removal (total removal—ABL, Figure 6) ranged from 0% (BP26DM, BPPH, BPFL and BPAF) to 71 ± 0.1% (BPC2 in co-culture). Statistically significant differences (*p* ≤ 0.05) between the three cultures were found for BPS, BPA, BPC2, BPC, BP26DM, BPM, BPP and BPPH. In the axenic/mixed cultures, microalgae removed more than 70% of BPC2 whereas bacteria removed less than 20% (Figure 6). In contrast, the differences in removal of BPPH between all three cultures were small (Figure 6) and followed the amount of biomass (Figure 5). In the case of BPS, BPC and BP26DM, the co-culture proved the most efficient; however, for BPS, the removal in all three cultures was lower than 20%. The highest removals were detected for BPC2 in the microalgal culture (71 ± 1%) and co-culture (71 ± 0.1%). The removals of BPA (22 ± 3%), BPM (28 ± 3%) and BPP (24 ± 2%) were higher in the bacterial culture than in the algal cultures.

Overall biotic removal (combined removal of all BPs) was, on average (white squares, Figure 7), highest in the co-culture (18 ± 18%), followed by the axenic culture (11 ± 15%) and lastly, the bacterial culture (10 ± 9%); however, differences between the three cultures series were not statistically significant (*p* > 0.05). The large standard deviations result from the broader range of biotic removal values of individual compounds.

## 3. Discussion

Due to their abundance, microalgae are considered important candidates for improving the resilience of the global ecosystem, which is at hazard due to pollution from industrial and urban waste. It is therefore vital to gather evidence that could indicate the mechanisms of their interaction with pollutants, which include also interactions with other microorganisms. As there is growing evidence that SCPs mediate the interactions between microalgae and bacteria, we addressed these issues for the first time in this work. We wished to see the ability of microalgae to remove BPs in the presence and absence of bacteria and seek possible connections between these activities.

Previous results on freshwater microalgae *C. vulgaris* and mixed culture of microorganisms from HRAP indicated that the removal is connected mainly to the number and diversity of microorganisms [34]. Based on these results we hypothesized that the co-culture of microalgae and bacteria would be superior in removal of BPs as both species would proliferate better, and we hinted at the role of SCPs in interaction between the species. Our hypothesis was confirmed as regards the overall biotic removal, as indeed, the mixed culture showed the best removal (Figure 6), which can be explained by a larger abundance of microorganisms in the mixed culture (Figure 1). However, the percent of the overall removal was not high, and all three cultures removed less than 20%, which is less than previously considered cultures with *C. vulgaris* or a mixture from wastewater facility HRAP [34]. More importantly, in contrast to previous studies, we found that the removal by *P. tricornutum* can vary greatly between the substances tested. *P. tricornutum* was shown to remove practically all (more than 99%) BPC2 either in axenic culture or in co-culture (Figure 5). Presence of bacteria increased the proliferation of the microalgae (Figure 1); however, we could not detect this in a further increase in removal, for the reason that it was already almost 100% in the axenic culture. Our results indicate that the overall removal of a mixture of pollutants can be increased by introducing in the culture the proper society of microorganisms that are specialized for the pollutants considered, which indicates that far greater overall removal can be achieved by a proper design based on the knowledge of mechanisms.

The bacteria that were inoculated for experiments performed in this work were taken from co-cultures of other microalgae that we are growing in our laboratory (*Tetraselmis chuii* and *Rhodella* sp.). As the removal of pollutants is ultimately intended for large-scale systems, we did not see a need or any other argument for highly specified microorganisms. However, we needed a stable culture and these bacteria were successfully growing for more than one year in the same conditions. For their identification, two different colony morphologies prevailed in the BD Columbia agar (Appendix A). Both were identified by 16S rRNA PCR to be members of Gram-negative *Alphaproteobacteria*, genus *Thalassospira* (the same percent identity for two species: *Thalassospira xiamenensis* and *Thalassospira permensis*) that are characteristic for oligotrophic waters of the East Mediterranean. The choice of bacteria proved to be good as they supported the proliferation of *P. tricornutum*.

As regards the role of SCPs we wished to identify them in the conditioned media of our system and better understand the mechanisms of their mediating role between the organisms. We have succeeded in harvesting SCPs from *P. tricornutum* in enough abundance to observe the major types of SCPs in proportions by electron microscopy, which has not yet been reported in this type of microalgae. This is a step forward with respect to previously published results on *P. tricornutum* [16], where only singular nanoalgosomes were presented. However, although we included the respective cultures from three parallels, the yield of harvested SCPs was not high enough for assessment of the differences in proteins in the samples (results not shown).

The field of SCPs is emerging, building on the notion that cells can exchange membrane-enclosed bits of their insides. Although the biophysical foundation of this interaction complies with theoretical predictions and extensive experimental data are accumulating, this new scientific field encounters many challenges. SCPs cannot be directly observed in their natural environment as they must first be harvested into samples. SCPs are not “small cells” that conserve their identity to some degree. The harvesting procedures and the characterization methods may transform, destroy or create new SCPs and strongly influence the identity of the samples. The yields of harvested SCPs are typically low which impedes research progress regarding analysis and applications [35,36]. It was estimated that a litre of conditioned culturing media of eukaryotic cells yields approximately 10^9^–10^11^ SCPs, an amount that is sufficient only for one single test [36]. As stated above, the challenge of low yield has not been overcome in our work. In general, due to small size and transient identity, SCP harvesting and characterization methods still do not have a golden standard, and any piece of evidence contributes to the developing field. We believe that attempts to better understand the role of SCPs in microorganism communities should be pursued.

### 3.1. Culture Growth

*P. tricornutum* grew exponentially throughout the experiment (Figure 1). The co-culture samples (Figure 1 EC, BC) reached a higher cell number density than the axenic ones (Figure 1 EA, BA), whereas the concentration of bacteria in both the co-culture and bacterial cultures reached the plateau phase of growth and declined towards the end of the experiment (Figure 1 EC, BC, EB, BB). In addition, the bacterial growth rate and the maximum cell number density were lower in the co-culture (Figure 1 EC, BC) compared to the bacterial cultures (Figure 1 EB, BB). The organic nutrients supplied with the LB can explain these growth curves and cell concentration limits. In the co-cultures and bacterial cultures, organic residues from the bacterial inoculum (0.25% working volume) and an additional 1% working volume of LB broth were added. This finding means that approximately four times more nutrients were available from the LB supplementation in the bacterial cultures (EB, BB) compared to the co-culture samples (EC, BC). This is proportional to the maximum bacteria concentration reached in these two sample types (3.6 ± 0.7 × 10^6^ mL^−1^ and 2.8 ± 0.2 × 10^6^ mL^−1^, in EC and BC, respectively, and 10.3 ± 0.4 × 10^6^ mL^−1^ and 10.6 ± 1.1 × 10^6^ mL^−1^, in EB and BB, respectively, Figure 1). *P. tricornutum* is known to have antimicrobial activity against various bacteria [37,38,39]. The higher microalgal cell number densities in the co-culture samples (Figure 1 EC, BC) were likely a consequence of the positive effect of bacteria on microalgal growth. Such effect was observed also in other systems; it was found that *Rhizobium* had a mutualistic effect on the growth of different types of green microalgae (i.e., *Chlamydomonas reinhardtii*, *C. vulgaris*, *Scenedesmus* sp. and *Botryococcis braunii*) where the growth rates of *Rhizobium* and microalgae increased from about 10 to 110% [40]. *C**. vulgaris* was found to be compatible with many heterotrophic bacteria [41,42], and with optimal inoculum the symbiotic co-culture proved to perform better in treatment of real municipal wastewater [41]. At the same time, microalgal and bacterial cells could be competing for nutrients from the small addition of a partially depleted LB medium during inoculation. In our experiments, no differences in growth rates (Figure 1) or cell morphology (Figure 2) were observed in cultures treated with BPs compared to the blank controls, suggesting that the concentrations of BPs used in the experiment were subtoxic to all microorganisms considered in this study.

### 3.2. Cell and SCP Morphology

The SEM imaging (Figure 2) of the bacterial samples (Figure 5 EC, BC, EB, BB) showed no noticeable difference or shift in the microbial community due to the presence of *P. tricornutum* (Figure 2). In addition, no zone of inhibition was observed between the microalgal and bacterial colonies on the solid media, indicating that *P. tricornutum* did not significantly affect bacterial growth in the cultures (Appendix A).

SEM revealed roughly spherical nanoparticles, 50–300 nm in size, in the isolates from all cultures containing *P. tricornutum*. These particles were homogenously sized (approx. 100 nm) and had a characteristically rough surface (Figure 4E). Cryo-TEM revealed particles ranging in size from tens to hundreds of nanometers enclosed by a membrane with a fibrillar coat. These particles were recognized as *P. tricornutum* EVs (Figure 4F, white arrowhead). However, Cryo-TEM also revealed other types of SCPs, such as electron-dense clusters (Figure 4F, black arrowhead). Therefore, it is not decisive what is the predominant type of SCPs in the culture and isolates.

It was previously reported that the production of EVs in cell lines could be increased upon different types of stress; hypoxia was shown to induce the EV release from breast cancer cell lines and mechanisms of this induction were studied [43]. The largest number of studies considered different types of cancer cells, followed by mesenchymal stem cells and cardiac cells [43]. In addition, mechanisms of EV formation in oxidative stress have been a subject of interest; it is indicated that EVs shed due to oxidative stress conditions can stimulate immune cells via toll-like receptors and cytokine release [44]. Moreover, the nutritional status of the cells is considered to be connected with the formation of EVs; it was shown that glucose depletion enhanced the formation of EVs in a rat myoblast H9C2 cell line [45]. It was suggested that the release of EVs subject to different stress stimuli can be viewed as a mechanism of homeostasis maintenance; however, the precise mechanisms of these responses remain to be elucidated [44].

### 3.3. Bisphenol Residue Mass Balance and Removal

Overall biotic BPs removal followed a similar trend as seen in previous studies [34], meaning that more residues of lipophilic (log P ≥ 6) compounds remained in the biomass phase whereas hydrophilic compounds (log P < 6) prevailed in the aqueous phase (Figure 5). Removals of BPS, BPC2, BPC, BP26DM and BPPH were higher in cultures with microalgal cells, whereas BPA, BPM and BPP removals were higher in the bacterial cultures. However, BPM, BPP and BPPH were mainly depleted in the abiotic controls. The co-culture was more efficient at overall removal of BPs than both the axenic microalgae and bacterial cultures only in the cases of BPC and BP26DM. Instead, different cultures were efficient in removing different BPs. Removals of BPC2 (in EA and EC) and BP26DM (in EC) were higher than 70%, constituting an appreciable fraction (Figure 6).

We found differences in removal between the cultures in 8 out of 18 BPs. We could not link these differences to log P, as the differences were present in BPS, which is a relatively hydrophilic compound (log P = 2.32), and in BPM, BPP and BPPH, which are relatively lipophilic (log P of 6.72, 6.72 and 7.34, respectively). Addressing the structure–activity relationships would require further experiments.

Compared to *C. vulgaris*, which, except for BPC2, reached more than 50% biotic removal after 144 h for 2, 4′-BPF, 4,4′-BPF, BPC, BP26DM and BPM [34], *P. tricornutum* was generally less efficient at BPs removal. Moreover, a mixed culture of diverse microalgae and bacteria from a high-rate algal pond grown in wastewater (HRAP) performed on average even more efficiently than the *C. vulgaris* culture [34]. However, in BPC2 specifically, the total removal efficiency was high (>99%) in the axenic microalgae and in co-culture (Figure 5). Although not as successful in overall removal of BPs as *C. vulgaris*, *P. tricornutum* may be important as a partner in microbe society for its high ability to remove a particular pollutant. A potential reason for conspicuous behaviour of BPC2 could be sought in its content, as it is the only chlorinated compound, which may influence its behavior. We speculate that *P. tricornutum* membranes (including SCPs) could play a role in the removal of a particular substance. Although small, membrane buds (Figure 2) and SCPs (Figure 4) present large surfaces with respect to the enclosed volume. Hypothetical enrichment of BPC2 in SCPs will be a subject of future work.

Microalgae-based technologies that rely on microalgae communities and aerobic heterotrophic microorganisms (primarily bacteria) are promising alternatives to conventional biological wastewater treatment. Although many resources are exploited linearly (extract–use–disposal), the reuse of resources by recycling wastewater is in line with circular economy principles. One of the main advantages of utilizing microalgae-based wastewater treatment is nutrient- and energy-rich microalgal biomass production and the production of reclaimed water. Notably, the valorization of biomass and reclaimed wastewater is in line with the principles of the EU’s Circular Economy Action Plan [46]. It has been shown that microalgae-based wastewater treatment technologies may exhibit CEC removal efficiency comparable or superior to conventional biological systems [47,48].

Understanding microalgal–bacterial interactions is crucial for effective microalgae-based wastewater treatment, production of biomass and value-added products. Mutualistic interactions include the exchange of macronutrients, where bacteria benefit from fixed carbon from the microalgae, while bacteria may fix nitrogen in exchange [49]. Furthermore, micronutrient exchange may also occur, notably in the form of vitamin B_12_ or B_1_ excretion by bacteria [49]. Signal transduction is another form of microalgal–bacterial interactions, where bacterial secretions may induce morphogenesis in microalgae, microalgal secretions inhibit bacterial quorum sensing or microalgae may inhibit the growth of bacteria and vice versa [50]. Lastly, evidence in the chloroplast genome of diatoms and dinoflagellates and diatom ornithine-urea cycle genes point to horizontal gene transfer between microalgae and bacteria as another type of interaction that shaped such partnerships through evolutionary processes [50]. The study of microalgal–bacterial interactions is also of significance to understanding the underlying mechanisms of ecology (mutualistic or host–pathogen interactions, spreading of resistance and disease) and presents a significant applicative potential for biotechnology and other industries [51].

## 4. Materials and Methods

Bacteria were identified by 16S rRNA PCR. Microorganism growth rates were determined by measuring cell concentration in culture samples using flow cytometry (FCM). Isolates of SCPs were prepared by differential centrifugation. Cultures and isolates were examined by Cryo-TEM and SEM to evaluate possible qualitative differences between samples. BPs were analyzed by gas GC-MS/MS. For one week, the cultures were grown in media supplemented with BPs, and blank cultures (the respective microorganisms in the media without added BPs) were grown as controls.

### 4.1. Cultures and Medium Composition

An axenic culture of the diatom *P. tricornutum* (CCAP 1052/1A, Culture Collection of Microalgae & Protozoa, Oban, Scotland) [16] was chosen as the model organism. Two bacterial samples, one with prevailing Gram-positive and one with prevailing Gram-negative unidentified bacilli, were taken from other in-house marine microalgal cultures to compose a bacterial culture for inclusion in the experiment. Microorganisms were cultured according to a modified procedure described in [17]: cultures were grown in 300 mL Erlenmeyer flasks on an orbital shaker (Vibromix 40, Domel, Železniki, Slovenia) at 130 RPM. Irradiance was provided with Osram Fluora (Germany) fluorescent lights, resulting in approximately 41 ± 6 μmol m^2^ s^−1^ of photosynthetically active radiation, with a photoperiod of 16:8 (light:dark).

Artificial marine water (MW) was prepared by dissolving 22 gL^−1^ of an artificial sea salt mix (Reef Crystals-Aquarium Systems, Sarrebourg, France) in distilled water. Guillard’s (F/2) marine water enrichment solution (ref. nr. G0154, Sigma Aldrich, St. Louis, MI, USA), and Luria-Bertani (LB) broth (ref. L3022, Sigma Aldrich, USA) were added to support microalgal and bacterial growth, respectively. In addition, MW-F/2 was prepared by adding 20 mL of enrichment solution per 1 L of MW. MW-LB was prepared by adding 20 g of LB per 1 L of MW. A solid MW-LB medium was prepared by adding 1% agar to MW-LB. All media were filtered through a 0.2 micron cellulose filter (ref. 11107-47-CAN, Sartorius Stedim Biotech GmbH, Gottingen, Germany) and autoclaved before use.

The microalgal cultures were inoculated into the inorganic medium of Guillard’s F/2-enriched seawater (MW-F/2). Given that the medium was autoclaved, the concentration of biotin, thiamine and vitamin B_12_ was likely reduced before inoculation. Two precultures of bacteria were prepared—one from a culture of *Rhodella* sp. and one from a culture of *Tetraselmis chuii*. Some LB broth (0.5% culture volume) was added to the co-cultures and bacterial cultures with the bacterial inoculum (already depleted in nutrients during the preculture). The medium for the bacterial cultures was additionally supplemented with 1% LB broth to ensure some bacterial growth. No additional LB broth was added to the co-cultures since the microalgae would provide organic support for bacterial growth. BPs were spiked into the experimental and abiotic control series by spiking 40 µL of a MeOH solution containing 5 mg/mL of each of the 18 BPs, resulting in a nominal concentration of 1 µg/L of each compound (Table 1). Individual compounds are identified in Appendix A.

For bacterial inoculum, several colonies of bacteria were transferred from solid (1% Agar, Sigma-Aldrich, USA, Lot BCCB5406 prepared in MW-LB) to 2 mL of liquid MW-LB medium (20 gL^−1^) and allowed to grow for three days at room temperature on a rotational shaker (HulaMixer, Termofisher Scientific, Waltham, MA, USA) at 10 rpm. Before the experiment, both precultures were mixed before inoculation. In the experiment, the medium for bacterial cultures was supplemented with 1% MW-LB to provide a necessary amount of nutrients, predicted to be comparable to that provided by the microalgae in the co-culture. Cultures for the experiment were prepared as presented in Table 1. The experiment lasted for 168 h, and samples were taken for measurements of cell concentration of microalgae and bacteria (0 h, 72 h and 168 h), BPs concentration (72 h and 168 h) and SCP isolation (168 h). This time interval was chosen on the basis of previous experience with *C. vulgaris*; it was found that the removal of most BPs increased during the first 150 h and then reached a plateau [52]. To confirm the absence of bacteria (contamination) in axenic microalgal and abiotic cultures, they were inoculated on MW-LB agar plates at the end of the experiment and incubated at room temperature for 14 days.

For identification, bacteria were cultured on BD Columbia agar containing 5% sheep blood (Becton Dickinson, Franklin Lakes, NJ, USA) (Appendix A) for five days at room temperature and 30 °C, respectively. Two prevailing colony morphotypes were analyzed by sequencing of the 16S ribosomal RNA gene according to the manufacturer’s instructions (Molzym GmbH, Bremen, Germany); the sequence of the 16S ribosomal RNA gene was visually corrected and edited and then compared with the public sequence database GenBank using the Nucleotide BLAST program run via the server of the National Centre for Biotechnology Information NCBI (Nucleotide BLAST: Search nucleotide databases using a nucleotide query (nih.gov)). The sequences of individual strains and a contig were analyzed. Sequences with a ≥99.0% match to a database sequence were considered to belong to the same species as the sequence with the highest similarity (Appendix A). In addition, all 16S ribosomal RNA gene sequences were compared to the RDP, Release 11 database (Ribosomal Database Project, RDP Release 11—Sequence Analysis Tools (msu.edu)), a highly curated database of aligned and annotated ribosomal RNA sequences and associated phylogenies. Each of the two variants was identified as *Thalassospira* sp.

### 4.2. Cell Concentration Measurement by Flow Cytometry

The cell concentration in the culture samples was determined by flow cytometry [52] using a MACSQuant Analyzer flow cytometer (Miltenyi Biotec, Bergisch Gladbach, Germany) and the related software. The following instrument settings were employed: FSC: 458 V; SSC: 467 V with a trigger set to 1.48, B3: 300 V; R1: 360 V. Non-fluorescent particles (NFPs), corresponding to bacteria, cell debris and SCPs, were detected from the forward (FSC) and side scatter parameter (SSC), as they were not autofluorescent. The microalgal cells were identified based on chlorophyll autofluorescence (AFP), detecting red emission (channels B3: 488 nm/655–730 nm, and R1: 635 nm/655–730 nm). Both AFP and NFP were quantified in all experimental and blank control series to check for possible contamination by microalgae or bacteria. Examples of AFP and NFP flow cytometer diagrams are shown in Figure 8.

### 4.3. SCP Isolation

SCPs were isolated by differential centrifugation according to the protocol commonly used for isolation of EVs as described in [17]. Cells and larger cell debris were cleared from the culture media in four centrifugation steps performed in a centrifuge Centric 400R (Domel, Železniki, Slovenia), using sterile polypropylene 15 mL conical centrifuge tubes 1: 660 g, 20 min, 4 °C; 2: 2640 g, 22 min, 4 °C; 3: 4000 g, 60 min, 4 °C; 4: 4000 g, 90 min, 4 °C. Finally, the SCPs remaining in the supernatant were pelleted by centrifugation at 118,000× *g*, 70 min, 4 °C (Ultracentrifuge Beckman L8-70M, rotor SW 55Ti, using thin open-top polypropylene tubes Ref. No. 326819 Beckman Coulter, Inc., Brea, CA, USA). The supernatant was removed, and the pellets were suspended in the remaining residual supernatant.

### 4.4. SEM Imaging

Samples were prepared for SEM by a protocol [53] adopted from [54]. Cultures and SCP isolates were applied on polycarbonate filter membranes (0.2 micron Isopore^TM^, ref. GTTP01300, Merk Millipore Ltd., Dublin, Ireland were used for culture samples, and 0.05 micron, ref. PCT00513100, Sterlitech, Auburn, WA, USA for NPs samples). Samples were then incubated for 2 h in 2% OsO_4_. The unbound osmium was removed by three steps of washing in distilled water and 10 min incubation was performed in each step before changing the solution. Then, samples were dehydrated in a graded series of ethanol (30%, 50%, 70%, 80%, 90%, absolute), treated with hexamethyldisilazane (30%, 50% mixtures with absolute ethanol, followed by pure hexamethyldisilazane), and air-dried. The samples were Au/Pd-coated (PECS Gatan 682) and examined using a JSM-6500F Field Emission Scanning Electron Microscope (JEOL Ltd., Tokyo, Japan).

### 4.5. Cryo-TEM Imaging

Samples of SCPs were prepared using Vitrobot Mark IV (Thermo Fisher Scientific, Waltham, MA, USA). Quantifoil^®®^ R 2/2, 200 mesh holey carbon grids (Quantifoil Micro Tools GmbH, Großlöbichau, Germany) were glow-discharged for 60 s at 20 mA and positive polarity in the air (GloQube^®®^ Plus, Quorum, Laughton, UK) [55]. Conditions were set at 4 °C, 100% relative humidity, blot time: 5 s, and blot force: 4. An amount of 2 µL of the sample with SCPs in suspension was applied to the grid, blotted, and vitrified in liquid ethane. Samples were visualized under cryogenic conditions using a 200 kV microscope Glacios with Falcon 3EC detector (Thermo Fisher Scientific, Waltham, MA, USA). The total electron dose was 30 eA^−2^.

### 4.6. Bisphenol Quantification

Sample preparation was conducted as described in Škufca et al. (2021) [56]. Briefly, samples were centrifuged at 6000× *g* for 20 min to separate the aqueous and biomass phases. A standard internal mixture (final concentration in the sample: 500 ngL^−1^ of BPAd_16_, ^13^C_12_-BPF, ^13^C_12_-BPS and ^13^C_12_-BPB each, 25 μL of 1 μg mL^−1^ solution) was also added, and the sample was filtered. The samples were then acidified and loaded onto MCX Prime (Waters, USA) solid-phase extraction (SPE) cartridges. The biomass phase was lyophilized and extracted with ACN/MeOH (80:20). Extracts were solvent-exchanged to 4.5 mL EtAc/Hex (25:75) and purified with Bond Elut Carbon/PSA (Agilent, Santa Clara, CA, USA) SPE cartridges.

BPs were analyzed using gas chromatography (GC, model 7890B, Agilent, USA) with tandem mass spectrometry (MS/MS, model 7000, Agilent, Santa Clara, CA, USA). Separation was achieved using a DB-5 MS capillary column (30 m × 0.25 mm × 0.25 μm; Agilent, Santa Clara, CA, USA) with helium as the carrier gas. Samples were injected in splitless mode at 270 °C. The compounds were ionized in electron impact (EI) mode at 70 eV and detected using multiple reaction monitoring modes (MRM). The total runtime was 24 min. Raw data on assessment of BPs are presented in Appendix A.

### 4.7. Data Analysis

Physicochemical properties of BPs (log P) were predicted based on compound structure, using the Marvin Suite (ChemAxon, Budapest, Hungary, https://chemaxon.com/). Data analysis and visualization were performed using the R programming language (R Foundation for Statistical Computing, Vienna, Austria, https://www.R-project.org/, version 4.1.2) in the R Studio environment (R Studio, Boston, MA, USA, https://www.rstudio.com/about/, version 2021.09.1). The software packages “tidyverse” [57], “xslx” [58] and “rstatix” [59] were used to analyze the data. The Kruskal–Wallis one-way analysis of variance was used to determine significant differences between series. BPs removal was calculated according to measured BPs mass in each phase (*m*_a_-aqueous and *m*_b_-biomass) at a given time according to the initial nominal spiked mass (*m*_0_ being 200 ng, equal to a spiked concentration of 1000 ng L^−1^ per compound). Removal was calculated as total removal using the following equation:(1)$$Removal_{T}\;\left(\%\right)=\frac{\left(m_{0}-\left(m_{a}+m_{b}\right)\right)}{m_{0}}\times 100\%$$

The biotic removal was calculated by subtracting the total removal in ABL series from the total removal of the experimental series:(2)$$Removal_{\mathrm{BIO}}\;\left(\%\right)=Removal_{T}-Removal_{\mathrm{ABL}}$$

## 5. Conclusions

We have observed notable differences in removals of some of the BPs between the axenic *P. tricornutum*, mixed and *Thalassospira* sp. cultures. Finding the respective organisms and creating highly efficient microbe societies could be key for the improvement of wastewater treatment. In particular, it is important to identify the bacteria strains to improve BPs biodegradation. Although the overall removal of BPs was found to be the greatest in the mixed culture, which was connected to greater biomass, *P. tricornutum* proved excellent in the removal of almost all BPC2. Our results indicate that some organisms may have the ability to remove specific pollutants with high efficiency. We have visualized numerous SCPs in all samples; however, the yield of SCPs was too low to give a decisive answer as regards their role in BPs removal. Attempts to better understand the role of different types of SCPs, either from algae or bacteria, in microorganism communities should be pursued.

## Figures and Tables

**Figure 1 ijms-23-08447-f001:**
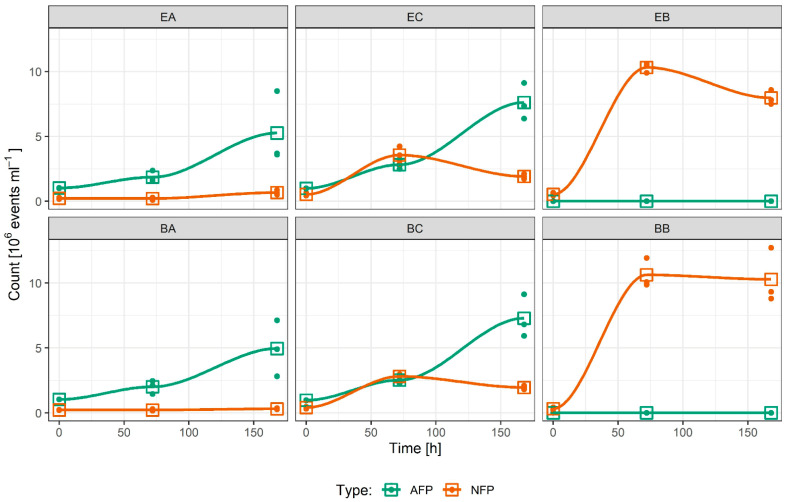
Flow cytometer scatter plots showing growth of cultures: experimental series (addition of BPs): axenic microalgae (EA), co-culture of microalgae and bacteria (EC) and bacterial culture (EB); blank control series (no addition of BPs): axenic microalgae (BA), co-culture of microalgae and bacteria (BC) and bacterial culture (BB). Microalgae are presented by autofluorescent events (AFPs) whereas bacteria and SCPs are presented by non-fluorescent events (NFPs). Single observations are shown as dots, whereas the mean value of three replicates is presented as an empty square.

**Figure 2 ijms-23-08447-f002:**
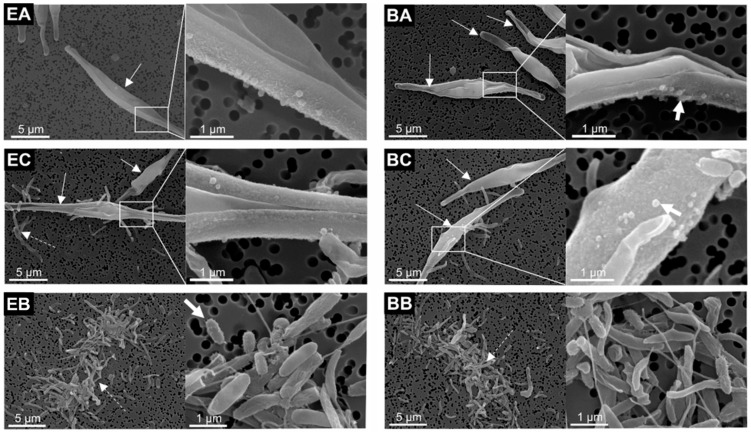
Scanning electron microscopy images of experimental (left side) and blank (right side) cultures of axenic microalgae (EA, BA, respectively), co-culture (EC, BC, respectively) and bacteria (EB, BB, respectively). Arrows point to fusiform cells (white arrows), bacteria (dashed white arrows) and protrusions on cell surface (fat white arrows). The insets in white rectangles are presented at higher magnification at the right side of the images.

**Figure 3 ijms-23-08447-f003:**
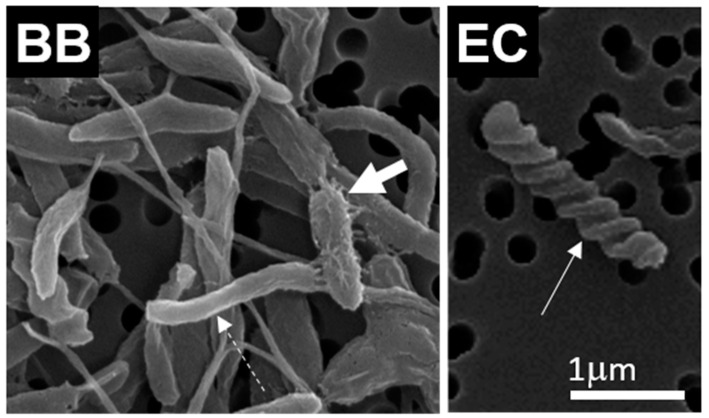
Close-up SEM images of examples of differently shaped bacteria found in bacterial cultures: BB (blank bacterial culture) and EC (experimental co-culture). Dashed white arrow points to rod-shaped cell, fat white arrow points to rod-shaped cell with a rough hairy surface and white arrow points to a drill-shaped cell.

**Figure 4 ijms-23-08447-f004:**
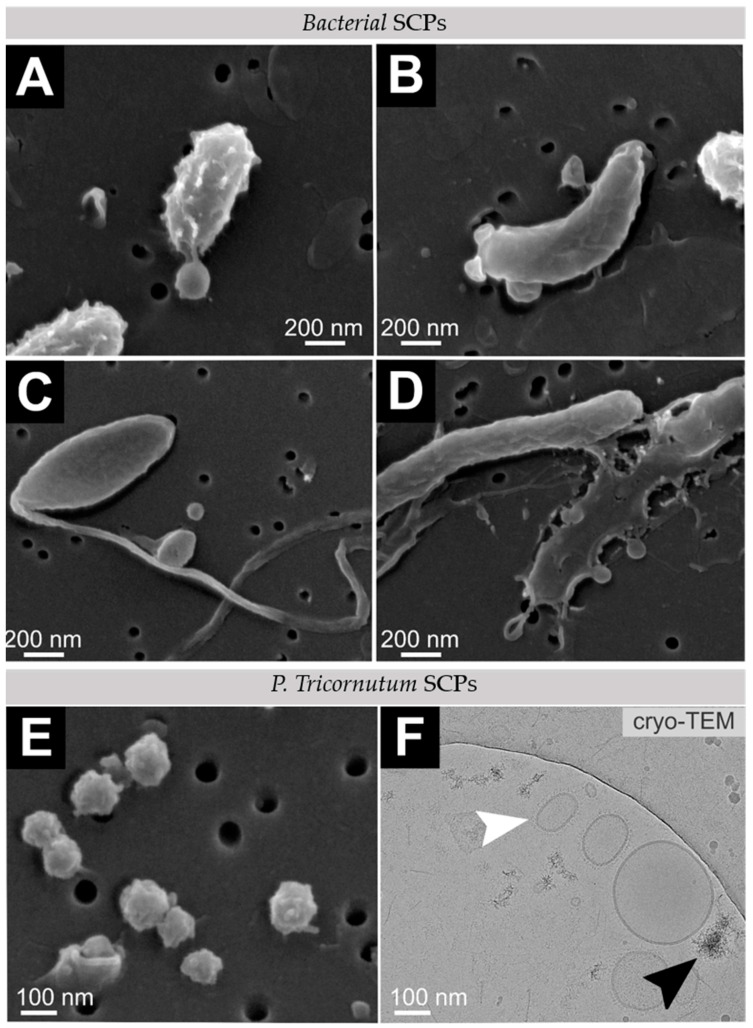
Electron microscopy imaging of osmium-fixed samples of *Thalassospira* sp. SCP isolates (**A**–**D**) and *P. tricornutum* SCP isolates (**E**), and Cryo-TEM image of *P. tricornutum* SCP isolate (**F**). The white arrowhead points to an EV, and the black arrowhead points to an electron-dense cluster.

**Figure 5 ijms-23-08447-f005:**
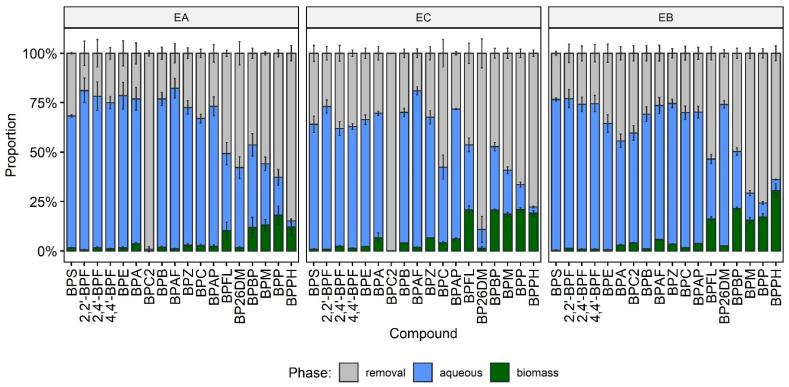
Proportion of BPs in axenic microalgae (EA), co-cultures (EC) and bacterial cultures (EB) 168 h after addition of BPs to the medium (*n* = 3 for each group). The proportions of the aqueous phase, biomass phase and calculated removal are shown with mean ± standard deviation. Compounds are arranged from lowest (BPS, log P = 2.3) to highest (BPPH, log P = 7.3) log P value.

**Figure 6 ijms-23-08447-f006:**
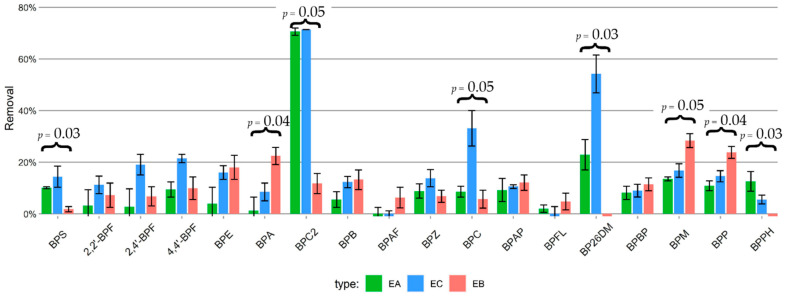
Biotic removal of BPs in axenic microalgae (EA), co-cultures (EC) and bacterial cultures (EB) 168 h after addition of BPs to the medium (*n* = 3 for each group). Significant differences are denoted by *p*-value of the Kruskal–Wallis test. Compounds are arranged from lowest (BPS, log P = 2.3) to highest (BPPH, log P = 7.3) log P value.

**Figure 7 ijms-23-08447-f007:**
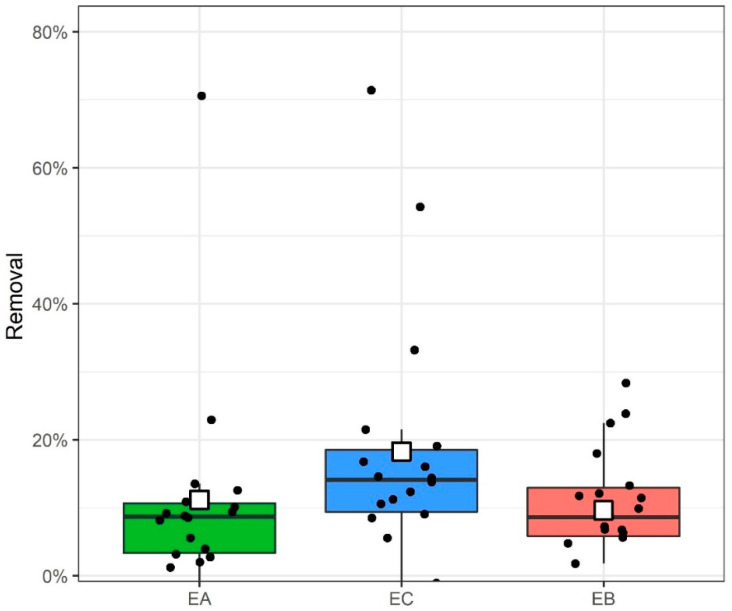
Overall biotic removal of all 18 BPs in axenic microalgae (EA), co-cultures (EC) and bacterial cultures (EB) 168 h after addition of BPs to the medium (*n* = 18 for each group). Boxplot explained: thick horizontal line is the median, box spans from the first to the third quartile, whereas whiskers span 1.5 interquartile range. White square is the mean of all observations. Points representing the removal of individual compounds are also shown to observe their distribution easier.

**Figure 8 ijms-23-08447-f008:**
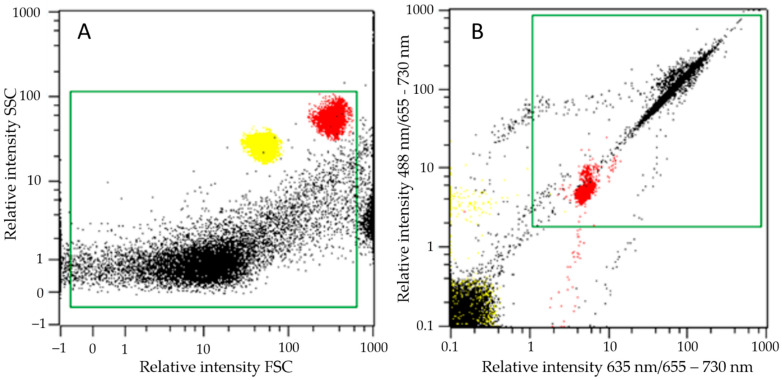
Flow cytometer plots of (**A**) NFP, attributed mainly to bacteria and SCPs; (**B**) AFP, attributed to microalgal cells. Clusters of fluorescent calibration beads are overlaid for comparison: 2 µm beads (left, yellow) and 3 µm beads (right, red). Yellow beads in B are clustered in the bottom left, with a negligible amount above that. Gates for selecting points for quantification of NFPs and AFPs (green squares) are also shown.

**Table 1 ijms-23-08447-t001:** Preparation of series used in the experiment.

	Series	r *	F/2 [mL]	LB [mL]	Microalgae Inoculum [mL]	Bacterial Inoculum [mL]	Bisphenol Standard [µL]	Blank MeOH [µL]	Ill **
control series	abiotic dark (ABD)	3	200	0	0	0	40	0	NO
	abiotic light (ABL)	3	200	0	0	0	40	0	YES
	blank axenic microalgae (BA)	3	150	0	50	0	0	40	YES
	blank co-culture (BC)	3	149.5	0	50	0.5	0	40	YES
	blank bacteria (BB)	3	197.5	2	0	0.5	0	40	YES
experimental series	axenic microalgae (EA)	3	150	0	50	0	40	0	YES
	co-culture (EC)	3	149.5	0	50	0.5	40	0	YES
	bacteria (EB)	3	197.5	2	0	0.5	40	0	YES

*—replicates of the same series, **—illumination with fluorescent tubes.

## Data Availability

Data may be obtained by contacting the authors.

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
