# Peer review of "Interaction between Microalgae P. tricornutum and Bacteria Thalassospira sp. for Removal of Bisphenols from Conditioned Media"

_ijms, 2022, doi:10.3390/ijms23158447_

Round 1

Reviewer 1 Report

The manuscript entitled „Indicated role of small cellular particles in mediating inter-species interaction for removal of bisphenols from conditioned media“ was submitted by David Škufca, Darja Božič and co-workers to the MDPI-journal “International Journal of Molecular Sciences” in order to be considered for publication as an “Article”.

The manuscript reports on cellular particles of Phaeodactylum tricornutum (serving as a model model organism) and their impact on the removal of fourteen different bisphenols (representing model compounds for contaminants of emerging concerns; however, they had no influence on protein expression) from medium/wastewater. Particular focus is directed towards the investigation of three different types of cultures, i.e., axenic culture of P. tricornutum microalgae, bacterial culture and a co-culture of both. The results document a higher growth of the microalgae co-cultured with bacteria in comparison to axenic microalgae. Eight of the fourteen studies bisphenols revealed different profiles in removing among the three culture set-ups. However, the mechanism that causes different profiles regarding growth of algae and removal of bisphenols remains to be elucidated.

Basically, the current manuscript represents a continuation of the previous work of the working group - which probably also justifies the self-citations. The manuscript generally fits the section "Biochemistry" of the journal "IJMS", however, some aspects (especially formal issues) should be revised by the authors in order to re-consider the manuscript.

The manuscript is written in an understandable way, the data are presented in a comprehensible way - although a combination of results section and discussion might be appropriate. The separation requires repeatedly scrolling back to the data in the tables and figures in the discussion part. However, this is the choice of the authors. Somewhat odd is the explicit inclusion of an "Appendix", although there is also a "Supplementary Materials". I would unify this and dispense with the appendix in the manuscript.

It seems a bit strange to say that "in this work" another publication is cited, namely reference [15] ... then it might not have been in this work, but just in the other study. Please revise.

The paragraph lines 124-132 does not report on “Results” but on “Materials and Methods” and therefore it should be replaced within the manuscript.

According to the guidelines of the journal, tables and figures should tables and figures should appear in the order in which they are mentioned. This enables reader-friendly reading of the manuscript.

Caption of Figure 5/6: Please also provide the abbreviations EA, EC, EB in the caption as done in the figures before. Caption of Figure 5: mean ± standard deviation of how many independent experiments (n=?)? Please provide the number of independent experiments in the caption of the figure.

Figure 5: Can the authors provide a (potential) reason for the conspicuous behavior of BCP2? The authors are kindly asked to comment on that.

Figure 4A would benefit from better resolution. Therefore, the authors are kindly asked to improve the resolution. Moreover, what data are shown? One single experiment or mean (how about SD and n=? then)?

Can the authors discuss in more detail on the lipophilicity – in order to derive a putative correlation. Moreover, as eight out of fourteen bisphenols showed a profile, the discussion of a potential structure-activity-relationships should be included in the manuscript.

Although the MDPI-publishing will do thorough editing, the authors are kindly requested to harmonize the formal presentation of the references. Moreover, formal errors and inconsistency should be revised, such as space (lines 47, 152, 153, Figure 3: P. tricornutum, 302, 308, space with “g” – section 4.3. vs. Figure 4, space with °C, in section 4.3. vs. elsewhere in the manuscript, consistently (no) space with % - section 4.4. vs. elsewhere in the manuscript), consistent use of punctuation (lines 80, 84, 104, 166, 191, 233, 358; lines 120-121 vs. 92-94, 96 vs. 360), typos (lines 91 postulated, 98 showed, 107 reported, 221 showed), use of in-italics (line 96, 360; 107 vs. 402, 503), explanation of non-popular abbreviation when using for the first time in the manuscript (lines 108: TSD, COD; 171, line 261), use of symbols (line 150, 467-468), consistent use of abbreviations (lines 149 vs. 158, 328, 447, 127 vs. 464, 468), case shift 373/ Table A1 IUPAC name, consistent style (line 396/399; /mL resp. /L vs. mL-1 resp. L-1) and Table 1 (* and ** different size), line 497 removal of % once. The list is by no means exhaustive, but should serve as a guide for authors to further improve the formal quality of their manuscript. Please revise the entire manuscript according to such issues.

Author Response

We thank the Editors for the message that our manuscript entitled “Indicated role of small cellular particles in mediating inter-species interaction for removal of bisphenols from conditioned media” by D. Škufca et al. can be considered for publication subject to improvements suggested by four Reviewers. Also we thank the Reviewers for valuable comments. Two Reviewers thought that it would be of interest to identify bacteria in cultures, therefore we have undertaken experiments to identify them and have added the results to the manuscript. The SDS-PAGE analysis presented in the previous version was not of sufficient quality according to three Reviewers, therefore we have removed it from the manuscript. Two Reviewers thought that the scope of the work was not described clearly. To meet the requirements of the Reviewers, we have rewritten the Abstract and the Conclusions and added text in the Discussion. Reviewer 4 suggested to better present the morphology of bacteria, therefore we inserted a new figure (Figure 3) showing a close-up on differently shapes cells.

We are thankful to Reviewer 1 for thorough reading of the manuscript and pointing to many negligent mistakes in presentation. We have made small changes throughout the manuscript to minimize the negligent errors. We have followed the suggestion of Reviewer 1 that the Appendix should be presented as a Supplementary material. Also we have made improvements in Figures 2, 4, S1 and S2. We have inserted six new references [35,36,52-55] to support the new text in the Discussion. According to the suggestions of the Reviewer 4 we added more detailed explanation of the methods used in this study to assure repeatability.

We hope that we have improved our work to reach the quality appropriate for publication. Kindly find below the point to point answers to the Reviewer’s comments. The changes made in the manuscript are highlighted in yellow. Also, the lines where the changes have been made in the manuscript are given in the point to point answers.

Reviewer 1

Q1. Somewhat odd is the explicit inclusion of an "Appendix", although there is also a "Supplementary Materials". I would unify this and dispense with the Appendix in the manuscript.

A1. We have moved the Appendix to the Supplementary material.

Q2. It seems a bit strange to say that "in this work" another publication is cited, namely reference [15] ... then it might not have been in this work, but just in the other study. Please revise.

A2. We have rephrased the sentence to: "We have focused on the study of SCPs of P. tricornutum for it was previously recognized as one of the species with appreciable yields of SCPs in isolates from microalgal conditioned media [16,17]." (Lines 67-69).

Q3. The paragraph lines 124-132 does not report on "Results" but on "Materials and Methods" and therefore it should be replaced within the manuscript.

A3. We have moved the paragraph to the Material and Methods (Lines 445-451).

Q4. According to the guidelines of the journal, tables and figures should tables and figures should appear in the order in which they are mentioned. This enables reader-friendly reading of the manuscript.

A4. To ensure the consecutive order of referring to the Figures we have rephrased the reference to FCM diagrams in the subsection Culture growth to “The AFPs and NFPs were defined by the respective regions of the FCM diagrams as described in Materials and Methods.” (Lines 146-147)

Q5. Caption of Figure 5/6: Please also provide the abbreviations EA, EC, EB in the caption as done in the figures before. Caption of Figure 5: mean ± standard deviation of how many independent experiments (n=?)? Please provide the number of independent experiments in the caption of the figure.

A5. We have provided the abbreviations and the number of independent experiments in the captions to Figures 4 and 5 (Lines 236-239 and 252-255)

Q6. Figure 5: Can the authors provide a (potential) reason for the conspicuous behavior of BCP2? The authors are kindly asked to comment on that.

A6: A potential reason for conspicuous behaviour of BPC2 could be sought in its content, as is the only chlorinated compound which may influence its behavior. We speculate that P. tricornutum membrane (including SCPs) could play a role in removal of a particular substance. Although small, membrane buds (Figure 2) and SCPs (Figure 4) present large surface with respect to the enclosed volume. Hypothetical enrichment of SCPs in BPC2 will be a subject of future work. We have inserted this text in Discussion (Lines 409-418).

Q7. Figure 4A would benefit from better resolution. Therefore, the authors are kindly asked to improve the resolution. Moreover, what data are shown? One single experiment or mean (how about SD and n=? then)?

A7. Given all three Reviewers' comments, we decided to omit Figure 4 of the old version and the respective texts.

Q8. Can the authors discuss in more detail on the lipophilicity – in order to derive a putative correlation. Moreover, as eight out of fourteen bisphenols showed a profile, the discussion of a potential structure-activity-relationships should be included in the manuscript.

A8: If the Reviewer is referring to Figure 6, where we showed the different behavior of eight out of eighteen (in total, we studied 18 and not 14 compounds) bisphenols, we find that eight bisphenols showed significant differences in removal between the cultures. However, we do not believe this is linked to log P, as the differences were present in BPS, which is a relatively hydrophilic compound (log P = 2.32), and in BPM, BPP and BPPH, which are relatively lipophilic (log P of 6.72, 6.72 and 7.34, respectively). Without further experiments, we cannot draw any conclusions on structure-activity relationships. We have added a paragraph addressing this issue in the Discussion (Lines 400-404):

We found differences in removal between the cultures in eight out of 18 BPs. We could not link these differences to log P, as the differences were present in BPS, which is a relatively hydrophilic compound (log P = 2.32), and in BPM, BPP and BPPH, which are relatively lipophilic (log P of 6.72, 6.72 and 7.34, respectively). Addressing the structure-activity relationships would require further experiments.

Q9. Although the MDPI-publishing will do thorough editing, the authors are kindly requested to harmonize the formal presentation of the references. Moreover, formal errors and inconsistency should be revised, such as space (lines 47, 152, 153, Figure 3: P. tricornutum, 302, 308, space with "g" – section 4.3. vs. Figure 4, space with °C, in section 4.3. vs. elsewhere in the manuscript, consistently (no) space with % - section 4.4. vs. elsewhere in the manuscript), consistent use of punctuation (lines 80, 84, 104, 166, 191, 233, 358; lines 120-121 vs. 92-94, 96 vs. 360), typos (lines 91 postulated, 98 showed, 107 reported, 221 showed), use of in-italics (line 96, 360; 107 vs. 402, 503), explanation of non-popular abbreviation when using for the first time in the manuscript (lines 108: TSD, COD; 171, line 261), use of symbols (line 150, 467-468), consistent use of abbreviations (lines 149 vs. 158, 328, 447, 127 vs. 464, 468), case shift 373/ Table A1 IUPAC name, consistent style (line 396/399; /mL resp. /L vs. mL-1 resp. L-1) and Table 1 (* and ** different size), line 497 removal of % once. The list is by no means exhaustive, but should serve as a guide for authors to further improve the formal quality of their manuscript. Please revise the entire manuscript according to such issues.

A9. We thank the reviewer for pointing to these mistakes. We have corrected the manuscript throughout to our best effort.  

Reviewer 2 Report

The authors investigated the efficiencies in removing various bisphenols by microalge Phaeodactylum tricornutum and bacteria in a laboratory setting. While it was shown that specific bisphenols could be removed in the microalge-bacteria co-culture, axenic microalge or bacteria culture, there are some issues related to the reproducibility and interpretation of results that the authors should further address.

Major comments:

1. In the title and main texts, the authors specifically mentioned that "small cellular particles" might have a role in bisphenols removal. However, throughout the texts, I could not see any data suggesting the association between small cellular particles and bisphenols removal. Just to give some examples, does any of the bisphenols induce, or increase, the production of small cellular particles? Or do the small cellular particles encapsulate any bisphenol molecules, as an indication of the relatedness?

2. In Figure 4, the authors tried to use coomassie blue-stained SDS-PAGE gels for comparing the protein profile of different samples. However, this was not even a qualitative comparison because there was basically no reference marker, or normalization, for the protein profiles in each lane. Simply based on the SDS-PAGE analysis, it is quite difficult to interpret that "no effects of BPs and co-cultures on the protein profiles", as stated at line 210-211. An optimal approach is to use mass spectrumetry for quantifying in protein level. Or it is also acceptable to quantify the normalized total protein amount based on the coomassie blue-stained SDS-PAGE gels using image anlaysis tools, e.g. ImageJ.  If the authors want to keep using the gel images, they need to properly interpret them and try to avoid over interpretation.

3. From line 401 to 403, the authors mentioned that "one with a prevailing unidentified gram-negative bacillus (isolated from a Rhodella sp. culture) and one with a prevailing gram-positive bacillus (isolated from Tetraselmis chuii culture), it may impose an issue of reproducibility becasue it is not clear how one can prepare the bacteria precultures from the two algal cultures if the readers want to repeat the experiments. From the EM images, it is difficult to tell whether there is one or multiple bacterial species. The authors should provide sufficient information for these precultures. For example, it should be described in detail reagarding the procedures for isolating the bacteria. The authors may also consider to sequence the 16S rRNA of some colonies to have some ideas of their identities.

Minor comments:

1. At line 143, thei authors first mentioned Figure 8 before any of the other figures, which may be improper. The authors may want to make it as the first supplementary figure and mention it here.

2. In Figure 2, there are two panels labeled as "EA" and the authors should correct the typo.

3. I would suggest to combine Figure A1 and A2 into one figure as they seem to present the same type of data.

4. In Figure 7, the authors should also explain what the little white sqaure represent.

5. At line 279 and 280, the authors used "V", presumably to represent "volume". It is recommended to avoid this abbreviation.

6. In section 4.3, "g" was used, while in section 4.7, "RCF" was used, represent the centrifugal force, it is recommended to make it consistent for the abbreviation.

7. In section 4.8, the authors should list the versions of the softwares for data analysis.

Author Response

We thank the Editors for the message that our manuscript entitled “Indicated role of small cellular particles in mediating inter-species interaction for removal of bisphenols from conditioned media” by D. Škufca et al. can be considered for publication subject to improvements suggested by four Reviewers. Also we thank the Reviewers for valuable comments. Two Reviewers thought that it would be of interest to identify bacteria in cultures, therefore we have undertaken experiments to identify them and have added the results to the manuscript. The SDS-PAGE analysis presented in the previous version was not of sufficient quality according to three Reviewers, therefore we have removed it from the manuscript. Two Reviewers thought that the scope of the work was not described clearly. To meet the requirements of the Reviewers, we have rewritten the Abstract and the Conclusions and added text in the Discussion. Reviewer 4 suggested to better present the morphology of bacteria, therefore we inserted a new figure (Figure 3) showing a close-up on differently shapes cells.

We are thankful to Reviewer 2 for thorough reading of the manuscript and pointing to negligent mistakes in presentation. We have made small changes throughout the manuscript to minimize the negligent errors. We have followed the suggestion of Reviewer 1 that the Appendix should be presented as a Supplementary material. Also we have made improvements in Figures 2, 4, S1 and S2. We have inserted six new references [35,36,52-55] to support the new text in the Discussion. According to the suggestions of the Reviewer 4 we added more detailed explanation of the methods used in this study to assure repeatability.

We hope that we have improved our work to reach the quality appropriate for publication. Kindly find below the point to point answers to the Reviewers’ comments. The changes made in the manuscript are highlighted in yellow. Also, the lines where the changes in the manuscript have been made are given in the point to point answers.

Reviewer 2

Q1. In the title and main texts, the authors specifically mentioned that "small cellular particles" might have a role in bisphenols removal. However, throughout the texts, I could not see any data suggesting the association between small cellular particles and bisphenols removal. Just to give some examples, does any of the bisphenols induce, or increase, the production of small cellular particles? Or do the small cellular particles encapsulate any bisphenol molecules, as an indication of the relatedness?

A1. To clarify design of our study and interpret the results we added the text below to the Discussion (Lines 306-330):

“As regards the role of SCPs we wished to identify them in the conditioned media of our system and better understand the mechanisms of their mediating role between the organisms. We have succeeded to harvest SCPs from P. tricornutum in enough abundance  to observe the major types of SCPs in proportions by electron microscopy, which has not yet been reported in this type of microalgae. This is a step forward with respect to previously published results on P. tricornutum [16] where only singular nanoalgosomes were presented. However, although we included the respective cul-tures from three parallels, the yield of harvested SCPs was not high enough for as-sessment of the differences in proteins in the samples (results not shown). 

The field of SCPs is emerging, building on the notion that cells can exchange membrane-enclosed bits of their insides. Although the biophysical foundation of this interaction complies with theoretical predictions and extensive experimental data are accumulating, this new scientific field encounters many challenges. SCPs cannot be directly observed in their natural environment as they must first be harvested into samples. SCPs are not “small cells” that conserve their identity to some degree. The harvesting procedures and the characterization methods may transform, destroy or create new SCPs and strongly influence the identity of the samples. The yields of harvested SCPs are typically low which impedes research progress regarding analysis and applications [35,36]. It was estimated that a litre of conditioned culturing media of eukaryotic cells yields approximately 109–1011 SCPs, an amount that is sufficient only for one single test [36]. As stated above, the challenge of low yield has not been overcome also in our work. In general, due to small size and transient identity, SCP harvesting and characterization methods still do not have a golden standard, and any piece of evidence contributes to the developing field. We believe that the attempts to better understand the role of SCPs in microorganism communities should be pursued.” 

Also we would be happy to see and present decisive answers on SCPs as the extracellular vesicle research society is hoping for the breakthrough to applications in many fields. Since in the last 15 years such event did not take place, we believe that also smaller steps such as the one found in this work (visualization) should be considered worthy of attention.

Q2. In Figure 4, the authors tried to use coomassie blue-stained SDS-PAGE gels for comparing the protein profile of different samples. However, this was not even a qualitative comparison because there was basically no reference marker, or normalization, for the protein profiles in each lane. Simply based on the SDS-PAGE analysis, it is quite difficult to interpret that "no effects of BPs and co-cultures on the protein profiles", as stated at line 210-211. An optimal approach is to use mass spectrumetry for quantifying in protein level. Or it is also acceptable to quantify the normalized total protein amount based on the coomassie blue-stained SDS-PAGE gels using image anlaysis tools, e.g. ImageJ.  If the authors want to keep using the gel images, they need to properly interpret them and try to avoid over interpretation.

A2. According to the comments of all three reviewers, we omitted Figure 4 of the old version and the respective texts.

Q3. From line 401 to 403, the authors mentioned that "one with a prevailing unidentified gram-negative bacillus (isolated from a Rhodella sp. culture) and one with a prevailing gram-positive bacillus (isolated from Tetraselmis chuii culture), it may impose an issue of reproducibility because it is not clear how one can prepare the bacteria precultures from the two algal cultures if the readers want to repeat the experiments. From the EM images, it is difficult to tell whether there is one or multiple bacterial species. The authors should provide sufficient information for these precultures. For example, it should be described in detail reagarding the procedures for isolating the bacteria. The authors may also consider to sequence the 16S rRNA of some colonies to have some ideas of their identities.

A3. According to the comments of Reviewers 2 and 3 we have performed the identification of bacteria. Two precultures of Thalassospira sp. were found – one with a prevailing unidentified gram-negative bacillus (isolated from a Rhodella sp. culture) and one with a prevailing gram-positive bacillus (isolated from Tetraselmis chuii culture). Thalassospira sp. was identified by sequencing the 16S ribosomal RNA gene according to the manufacturer's instructions (Molzym GmbH, Bremen Germany). The sequence of the 16S ribosomal RNA gene was also analysed in Nucleotide BLAST® (National Center for Biotechnology Information, Maryland, USA) and in RDP (Ribosomal Database Project), a highly curated database of aligned and annotated rRNA sequences and associated phylogenies. In both cases, there was a 100% match to the genus Thalassospira. The species level was not reliably identified.

We inserted these results into Methods, Lines 487-496.

Minor comments:

Q4. At line 143, the authors first mentioned Figure 8 before any of the other figures, which may be improper. The authors may want to make it as the first supplementary figure and mention it here.

A4. We rephrased the referencing (Lines 146 and 147).

Q5. In Figure 2, there are two panels labeled as "EA" and the authors should correct the typo.

A5. We have corrected the mistake, thanks to the Reviewer.

Q6. I would suggest to combine Figure A1 and A2 into one figure as they seem to present the same type of data.

A6. We have combined figures as suggested, we kept only the larger magnification.

Q7. In Figure 7, the authors should also explain what the little white sqaure represent.

A7. We have added the explanation: "white square is the mean of all observations". (Line 265).

Q8. At line 279 and 280, the authors used "V", presumably to represent "volume". It is recommended to avoid this abbreviation.

A8. We have replaced the symbol V with the word "volume" in both places (Lines 341 and 342).

Q9. In section 4.3, "g" was used, while in section 4.7, "RCF" was used, represent the centrifugal force, it is recommended to make it consistent for the abbreviation.

A9. We have replaced "RCF" with a multiplicity of g (Line 562).

Q10. In section 4.8, the authors should list the versions of the softwares for data analysis.

A10. We have provided the information on the version of the software (Lines 578,579).

Reviewer 3 Report

The manuscript "Indicated role of small cellular particles in mediating inter-species interaction for removal of bisphenols from conditioned media" by Škufca D., Božič D. et al. is devoted to the biodegradation of bisphenols.

The study  design is not clear from the presented manuscript:

1. Which bacterial species was used in the study? The authors don't characterize the bacterial culture at all. As well as we don't understand why the culture had been used in the experiment.

2. The results of the presented experiments are extremely unclear sometimes. For example, section 2.2:

"We observed no effects of BPs and co-cultures on the protein profiles (bands obtained by SDS-PAGE) in any of the three cultures: microalgal, bacterial, or co-culture (Figure 4). The amount of proteins in the isolates was too low to allow comparison between samples, as bands were not visible after staining."

It's not clear why the authors could not to isolate enough protein from samples? Why they use SDS-PAGE for protein profiling? How could we compare the proteins isolated from bacteria and microalgae by SDS-PAGE data?

3. The reviewer has a lot of questions about every part of the work due to the experiment design as well as the interpretation of results are exteremely far from correct.

Guess, the manuscript could not be recognised as a finished work with clear results and should be rejected in the current form.

Author Response

We thank the Editors for the message that our manuscript entitled “Indicated role of small cellular particles in mediating inter-species interaction for removal of bisphenols from conditioned media” by D. Škufca et al. can be considered for publication subject to improvements suggested by four Reviewers. Also we thank the Reviewers for valuable comments. Two Reviewers thought that it would be of interest to identify bacteria in cultures, therefore we have undertaken experiments to identify them and have added the results to the manuscript. The SDS-PAGE analysis presented in the previous version was not of sufficient quality according to three Reviewers, therefore we have removed it from the manuscript. Two Reviewers thought that the scope of the work was not described clearly. To meet the requirements of the Reviewers, we have rewritten the Abstract and the Conclusions and added text in the Discussion. Reviewer 4 suggested to better present the morphology of bacteria, therefore we inserted a new figure (Figure 3) showing a close-up on differently shapes cells.
We have made small changes throughout the manuscript to minimize the negligent errors. We have followed the suggestion of Reviewer 1 that the Appendix should be presented as a Supplementary material. Also we have made improvements in Figures 2, 4, S1 and S2. We have inserted six new references [35,36,52-55] to support the new text in the Discussion. According to the suggestions of the Reviewer 4 we added more detailed explanation of the methods used in this study to assure repeatability.
We hope that we have improved our work to reach the quality appropriate for publication. Kindly find below the point to point answers to the Reviewers’ comments. The changes made in the manuscript are highlighted in yellow. Also, the lines where the changes in the manuscript have been made are given in the point to point answers.

Reviewer 3

The manuscript "Indicated role of small cellular particles in mediating inter-species interaction for removal of bisphenols from conditioned media" by Škufca D., Božič D. et al. is devoted to the biodegradation of bisphenols.

Q1. The study  design is not clear from the presented manuscript:

A1. We tried to express ourselves better. We added a paragraph at the end of Introduction stating our aims (Lines 120-134). Also we have changed the Abstract, added text at the beginning of the Discussion (Lines 267-330) and rewrote the Conclusions (Lines 592-601) as given below:

Abstract (Results and Conclusions)

The microorganism growth rate was determined by flow cytometry. Cultures and SCP isolates were imaged by scanning electron microscopy (SEM) and cryogenic transmis-sion electron microscopy (Cryo-TEM), BPs were analyzed by gas chromatography cou-pled with tandem mass spectrometry (GC-MS/MS). The overall removal of BPs in axenic P. tricornutum, mixed and bacterial cultures was 11%, 18% and 10%, respec-tively. Notable differences in removal of eight out of 18 BPs between the axenic, mixed and bacterial cultures were found. Numerous SCPs from all three cultures were de-tected by electron microscopy. P. tricornutum in axenic culture and in mixed culture removed almost all BPC2. Our results indicate that some organisms may have ability to remove specific pollutant with high efficiency. Finding the respective organisms and creating microbe societies seems to be key for improvement of wastewater treatment. Further research on the mechanisms of interspecies communication is needed to ad-vance the understanding of microbial communities at the nano-level. The study of the role of SCPs in microorganism communities should be pursued. 

Introduction

In this work, we attempted to elucidate the mechanisms underlaying the removal of BPs by micoroalgae. Based on previous results [26,32] we hypothesized that a co-culture of microalgae and bacteria would be more efficient in removing BPs than a pure axenic microalgal or bacterial culture, due to greater expected biomass and potential mutualism in a mixed culture. Three different in vitro cultures were studied: (1) an axenic microalgal culture of P. tricornutum, (2) a co-culture of microalgae and bacteria, and (3) a bacterial cul-ture. We wished to see the ability of microalgae to remove pollutants in the presence of bacteria and without bacteria and seek possible connections between these activities. As a possible underlaying mechanism we considered the mediated interaction between the microorganisms by SCPs. To fulfil our goals, we followed the abundance of micro-organisms in the culture during growth and determined the concentrations of BPs in the medium. We isolated SCPs from the conditioned media and observed them by SEM and Cryo-TEM. The study aims to connect the role of SCPs s as mediators of intercellu-lar interactions to the broader field of ecology. To our best knowledge, in this work the above issues were addressed for the first time.

Discussion

Due to their abundance, microalgae are considered important candidates for the resiliation of the global ecosystem that is at hazard due to pollution from the industrial and urban waste. It is therefore vital to gather evidence that could indicate the mecha-nisms of their interaction with pollutants which include also interactions with other microorganisms. As there is growing evidence that SCPs mediate the interactions be-tween microalgae and bacteria, we addressed these issues for the first time in this work. We wished to see the ability of microalgae to remove BPs in the presence and absence of bacteria and seek possible connections between these activities.

Previous results on freshwater microalgae C. vulgaris and mixed culture of micro-organisms from HRAP indicated that the removal is connected mainly to the number and diversity of microorganisms [34]. Based on these results we hypothesized that the co-culture of microalgae and bacteria would be superior in removal of BPs as both species would proliferate better, and we hinted at the role of SCPs in interaction between the species. Our hypothesis was confirmed as regards the overall biotic removal, as in-deed, the mixed culture showed the best removal (Figure 6) which can be explained by larger abundance of microorganisms in the mixed culture (Figure 1). However, the percent of the overall removal was not high, all three cultures removed less than 20%, which is less than previously considered cultures with C. vulgaris or a mixture from wastewater facility HRAP [34]. More importantly, in contrast to previous studies, we found that the removal by P. tricornutum can vary greatly between the substances tested. P. tricornutum was shown to remove practically all BPC2 either in axenic culture or in co-culture (Figure 5). Presence of bacteria increased the proliferation of the microalgae (Figure 1), however, we could not detect this in a further increase of removal, also for the reason that it was already almost 100% in the axenic culture. Our results indicate that the overall removal of a mixture of pollutants can be increased by introducing in the culture the proper society of microorganisms that are specialized for the pollutants considered, which indicates that far greater overall removal can be achieved by a proper design based on the knowledge of mechanisms. 

The bacteria that were inoculated for experiments performed in this work were taken from co-cultures of other microalgae that we are growing in our laboratory. As the removal of pollutants is ultimately intended for large scale systems, we did not see a need or any other argument for highly specified microorganisms. Furthermore, these bacteria were successfully growing for more than one year in the same conditions. Six different types of bacteria were found in the culture (not shown). The two most abun-dant ones were identified by 16S rRNA PCR to be members of gram negative Alhapro-teobacteria, family Thalassospira (the same % of two species: Thalassospira xiamenensis and Thalassospira permensis) that are characteristic for oligotrophic waters of East Med-iteran. The choice of bacteria proved to be good as they supported the proliferation of P. tricornutum.

As regards the role of SCPs we wished to identify them in the conditioned media of our system and better understand the mechanisms of their mediating role between the organisms. We have succeeded to harvest SCPs from P. tricornutum in enough ab-bundance to observe the major types of SCPs in proportions by electron microscopy, which has not yet been reported in this type of microalgae. This is a step forward with respect to previously published results on P. tricornutum [16] where only singular nanoalgosomes were presented. However, although we included the respective cul-tures from three parallels, the yield of harvested SCPs was not high enough for as-sessment of the differences in proteins in the samples (results not shown). 

The field of SCPs is emerging, building on the notion that cells can exchange membrane-enclosed bits of their insides. Although the biophysical foundation of this interaction complies with theoretical predictions and extensive experimental data are accumulating, this new scientific field encounters many challenges. SCPs cannot be directly observed in their natural environment as they must first be harvested into samples. SCPs are not “small cells” that conserve their identity to some degree. The harvesting procedures and the characterization methods may transform, destroy or create new SCPs and strongly influence the identity of the samples. The yields of har-vested SCPs are typically low which impedes research progress regarding analysis and applications [35,36]. It was estimated that a litre of conditioned culturing media of eukaryotic cells yields approximately 109– 1011 SCPs, an amount that is sufficient only for one single test [36]. As stated above, the challenge of low yield has not been over-come also in our work. In general, due to small size and transient identity, SCP har-vesting and characterization methods still do not have a golden standard, and any piece of evidence contributes to the developing field. We believe that the attempts to better understand the role of SCPs in microorganism communities should be pursued. 

Conclusions

We have observed notable differences in removal of some of the BPs between the axenic P. tricornutum, mixed and bacterial cultures. While the overall removal of BPs was found to be the greatest in the mixed culture which was connected to greater bio-mass, P. tricornutum proved excellent in removal of BPC2. We have visualized numer-ous SCPs in all samples, however, the yield of SCPs was too low to enable further characterization.  Our results indicate that some organisms may have ability to re-move specific pollutant with high efficiency. Finding the respective organisms and creating highly efficient microbe societies could be key for improvement of wastewater treatment. In that, we believe that the attempts to better understand the role of SCPs in microorganism communities should be pursued. 

Q2. Which bacterial species was used in the study? The authors don't characterize the bacterial culture at all.

A2. According to the comments of Reviewers 2 and 3 we have performed test to identify bacteria. Two precultures of Thalassospira sp. were found – one with a prevailing unidentified gram-negative bacillus (isolated from a Rhodella sp. culture) and one with a prevailing gram-positive bacillus (isolated from Tetraselmis chuii culture). Thalassospira sp. was identified by sequencing the 16S ribosomal RNA gene according to the manufacturer's instructions (Molzym GmbH, Bremen Germany). The sequence of the 16S ribosomal RNA gene was also analysed in Nucleotide BLAST® (National Center for Biotechnology Information, Maryland, USA) and in RDP (Ribosomal Database Project), a highly curated database of aligned and annotated rRNA sequences and associated phylogenies. In both cases, there was a 100% match to the genus Thalassospira. The species level was not reliably identified.

We inserted these results into Methods, Lines 487-496.

Q3. As well as we don't understand why the culture had been used in the experiment.

A3. In our previous work [34] we have observed that the mixed cultures were more efficient in pollutant removal. Also the results of this work contribute to explanation of this observation. Namely, organisms have specific ability to remove a particular substance, therefore, pollutants consisting of several different substances will more likely be removed by the mixture of microorganisms, each removing the respective one. Furthermore, it is our particular interest to study cross-organism interactions by SCPs as there are indications that these are very fundamental and therefore important in all living systems.  

Q4. The results of the presented experiments are extremely unclear sometimes. For example, section 2.2:

"We observed no effects of BPs and co-cultures on the protein profiles (bands obtained by SDS-PAGE) in any of the three cultures: microalgal, bacterial, or co-culture (Figure 4). The amount of proteins in the isolates was too low to allow comparison between samples, as bands were not visible after staining."

A4. According to the comments, we have decided to omit Figure 4 from the old version and related text.

Q5. It's not clear why the authors could not to isolate enough protein from samples?

A5. As stated in the Discussion: The emerging field of SCPs encounters many challenges, one of them being low harvesting yield [35,36]. It was estimated that a litre of conditioned culturing media of eukaryotic cells yields approximately 109–1011 SCPs, an amount that is sufficient only for one single test [36]. The challenge of low yield has not been overcome also in our work. This issue is addressed in Discussion (Lines 306-330).  However, a step forward in study of microalgal SCPs was made as enough material was obtained to image numerous SCPs from P. tricornutum (Figures 2,4 and Supplementary material, Figure S1). Previous images revealed only singular SCPs [16].  

Q6. Why they use SDS-PAGE for protein profiling? How could we compare the proteins isolated from bacteria and microalgae by SDS-PAGE data?

A6. In agreement with the reviewers' comments on the SDS PAGE data, we omitted these results from the manuscript. We kept a sentence on the protein assessment in Lines 312-314: Although we included the respective cultures from three parallels, the yield of harvested SCPs was not high enough for assessment of the differences in proteins in the samples (results not shown). 

Q7. The Reviewer has a lot of questions about every part of the work due to the experiment design as well as the interpretation of results are exteremely far from correct. Guess, the manuscript could not be recognized as a finished work with clear results and should be rejected in the current form.

A7. Also we would be happy to present decisive answers, in particular in the field of extracellular vesicles where the whole extracellular vesicle society is hoping for the breakthrough to applications in many fields. As in the last 15 years such event did not take place, we believe that also smaller steps should be considered worthy of attention. We hope we have improved the clarity of presentation and would be happy to further explain the design and results of our work if necessary.

Reviewer 4 Report

 Title of Manuscript: Indicated role of small cellular particles in mediating  inter-species interaction  for removal of bisphenols from conditioned media. 

Manuscript Evaluation

The current work, in which single cultures of microalgae, bacteria and co-culture of both were applied, seems to be very crucial and significant in terms of stydy connection between removal of bisphenols representing CEC and environment of model bioreactor. Authors a lot of method of huge importance for such experiments. But clarity can be improved (see author’s comments).  Generally it is great job in preparing this article.  

Comments to the Author(s) to the manuscript  

Page 4- For Figure 1. Key (legend) below this figure should be definitely  changed cause now it is  not clearly visible when analysing the data shown  it hampers analysis of it.additionally abbreviations of EA, EC, EB and BA, BC, BB should be recalled in the heading for this graph.

Page 5- In Figure 2- left-hand side lower panel – EA should be changed into EB- nowi t is incorrect labelling of this pnale that indeed concerns bacterial culture with BPs. Moreover on the all panels arrows should beadded indicating fusiform shape of algae, bacteria  and drill-shaped cells. May be author shoud think about higher magnification of the micrographs , especially foru upper panels.

Similarly arrows should indicate for EA panel indicating SCPs.

Page 10- line 288- it is superficial statement that ,, ..positive effect of bacteria on microalgial growth, as observed in previous study [37-39}. Can authors explain it more in discussion in order to make discussion more valuable?

 Page 10- line 308- in which cel lines?- authors should complete it

Page 12- in materials and methods- can author explan why such times were  chosen for all the investigations? In what basis?

Material and methods_ for all the methods (there is only in one citaion given) authors should reffer to some literature

Page 13- 4.4 chapter- why glutaraldehyde was omitted in the method? Why 2% OSO4 was applied? What was the buffer used  for fixative?

Author Response

We thank the Editors for the message that our manuscript entitled “Indicated role of small cellular particles in mediating inter-species interaction for removal of bisphenols from conditioned media” by D. Škufca et al. can be considered for publication subject to improvements suggested by four Reviewers. Also we thank the Reviewers for valuable comments. Two Reviewers thought that it would be of interest to identify bacteria in cultures, therefore we have undertaken experiments to identify them and have added the results to the manuscript. The SDS-PAGE analysis presented in the previous version was not of sufficient quality according to three Reviewers, therefore we have removed it from the manuscript. Two Reviewers thought that the scope of the work was not described clearly. To meet the requirements of the Reviewers, we have rewritten the Abstract and the Conclusions and added text in the Discussion. Reviewer 4 suggested to better present the morphology of bacteria, therefore we inserted a new figure (Figure 3) showing a close-up on differently shapes cells.
We have made small changes throughout the manuscript to minimize the negligent errors. We have followed the suggestion of Reviewer 1 that the Appendix should be presented as a Supplementary material. Also we have made improvements in Figures 2, 4, S1 and S2. We have inserted six new references [35,36,52-55] to support the new text in the Discussion. According to the suggestions of the Reviewer 4 we added more detailed explanation of the methods used in this study to assure repeatability.
We hope that we have improved our work to reach the quality appropriate for publication. Kindly find below the point to point answers to the Reviewers’ comments. The changes made in the manuscript are highlighted in yellow. Also, the lines where the changes in the manuscript have been made are given in the point to point answers.

Reviewer 4

Q1. Page 4- For Figure 1. Key (legend) below this figure should be definitely  changed cause now it is  not clearly visible when analysing the data shown  it hampers analysis of it.additionally abbreviations of EA, EC, EB and BA, BC, BB should be recalled in the heading for this graph.

A1. We have improved the caption to Figure 1: we have clarified the text and explained the abbreviations pertaining to the cultures analysed.

Q2. Page 5- In Figure 2- left-hand side lower panel – EA should be changed into EB- nowi t is incorrect labelling of this pnale that indeed concerns bacterial culture with BPs. Moreover on the all panels arrows should beadded indicating fusiform shape of algae, bacteria  and drill-shaped cells. May be author shoud think about higher magnification of the micrographs , especially foru upper panels. Similarly arrows should indicate for EA panel indicating SCPs.

A2. We have improved Figure 2: we have corrected the negligent mistake (we thank the reviewer for pointing to it); we have inserted arrows pointing to fusiform cells (white arrows), bacteria (dashed white arrows) and protrusions on cell surface (fat white arrows). We have improved the caption to Figure 2: it now states that the insets in white rectangles are presented at higher magnification at the right side of the images. The presence of SCPs in Figure 2 is not decisive, rather, protrusions are observed on the cell surface (fat white arrows). To better show the different shapes of bacteria in samples we inserted a new figure (Figure 3) with higher magnification. We have equipped the figure with arrows pointing to different types of bacteria.    

Q3. Page 10- line 288- it is superficial statement that ,, ..positive effect of bacteria on microalgial growth, as observed in previous study [37-39}. Can authors explain it more in discussion in order to make discussion more valuable?

A3. We have expanded the Discussion as follows:

The higher microalgal cell number densities in the co-culture samples (Figure 1 EC, BC) are likely a consequence of the positive effect of bacteria on microalgal growth. Such effect was observed also in other systems; it was found that Rhizobium had mutu-alistic effect on the growth of different types of green microalgae (i.e. Chlamydomonas reinhardtii, C. vulgaris, Scenedesmus sp. and Botryococcis braunii) where the growth rates of Rhizobium and microalgae increased from about 10% to 110% [40]. C. vulgaris was found to be compatible with many heterotrophic bacteria [41,42], and at optimal inoculum, the symbiotic co-culture proved to perform better in treatment of real municipal wastewater [41]. At the same time, microalgal and bacterial cells could be competing for nutrients from the small addition of a partially depleted LB medium during inoculation. (Lines 350-358)

Q4. Page 10- line 308- in which cel lines?- authors should complete it

A4. We have expanded the discussionas follows:  

It was previously reported that the production of EVs in cell lines could be increased upon different types of stress; for example, hypoxia was shown to induce the EV release from breast cancer cell lines and mechanisms of this induction were studied [43]. The largest number of studies considered different types of cancer cells, followed by mesenchymal stem cells and cardiac cells [43]; it was found that EVs shed due to oxidative stress could stimulate immune cells via toll-like receptors and cytokine release [44] and nutritional stress caused by glucose depletion enhanced formation of EVs as demonstrated in rat myoblast H9C2 cell line [45]. It was suggested that the release of EVs subject to different stress stimuli can be viewed as a mechanism of homeostasis maintenance, however, the precise mechanisms of these responses remain to be elucidated [44]. (Lines 377-388)

Q5. Page 12- in materials and methods- can author explan why such times were  chosen for all the investigations? In what basis?

A4. We had previous experience with effect of BPs on microalgae (refs. [32,48,56]). It was found that the removal in most BPs increased during first 150 hours and then reached a plateau. These results were obtained in C. vulgaris [56] and it was assumed that similar time of observation (i.e. 7 days) would be relevant also in P. tricornutum. (Lines 502-504).

Q6. Material and methods_ for all the methods (there is only in one citaion given) authors should reffer to some literature

A6. We have cited references for culturing microalgae [16], [17], isolation of SCPs [17], flow cytometry (new reference [53]), preparation of samples for SEM (new reference [54]), and for Cryo-TEM (new reference [55]). Methods for determination of BPs [56] and for analysis of data [57]-[62] were already referenced.

Q7. Page 13 - 4.4 chapter- why glutaraldehyde was omitted in the method? Why 2% OSO4 was applied?

A7. One of techniques for preparation of samples for SEM that is widely practiced consists of fixation with glutaraldehyde to react with proteins and osmium tetroxide and uranyl acetate to increase the overall contrast and further stabilize membranes. In the standard protocols for SEM, specimens are fixed, dried, and sputtered with heavy metal. Alternatively, conductive staining is applied, based on binding of ligands in combination with osmium tetroxide to unsaturated lipids [54]. As our estimations of protein content of SCPs indicated that the samples are poor with proteins we considered fixation with osmium tetroxide more appropriate than the standard preparation based on fixation of proteins with glutaraldehyde. Preparation of SCPs for SEM is demanding as SCPs are very small and easily subjected to shape change. It is rarely used in characterization of SCPs, probably because the results of imaging after a day of preparation of the sample can be rather disappointing. However, if successful (if SCPs are not destroyed or lost during the preparation), SEM is presently the best technique that can reveal three dimensional shape of SCPs. Also, the frame of view that can be captured by different magnifications is considerably larger than that of Cryo TEM which renders the images more representative for the sample. Based on our experience with erythrocyte SCPs [54] where we were able to obtain SEM images rich with SCPs with the osmium preparation, we chose this preparation also for SCPs isolated from microalgal and bacterial cultures.

Q8. What was the buffer used for fixative?

A8. We did not use additional buffer for fixation of samples.

Round 2

Reviewer 1 Report

The authors D. Skufca, D. Bozic and co-workers submitted a revised version of their manuscript “Indicated role of small cellular particles in mediating inter-species interaction for removal of bisphenols from conditioned media” to the journal IJMS.

In their revision, the authors addressed every suggestion and concern mentioned before. They have implemented formal aspects (unification of Appendix and Supplementary Materials, removal to the paragraph Materials and Methods, order and anchoring of figures/tables in the text), clarified or even more clearly formulated previously inaccurately expressed statements in some places, given more precise details of the experimental procedures, and discussed the inconclusive matters as far as possible on the available data. The critical results of the SDS-PAGE analysis, however, have been completely removed by the authors instead of being corrected. Formal and linguistic inconsistencies were also largely corrected.

Overall, I suggest further processing of the manuscript. All the best!

Author Response

We thank the reviewer for valuable comments that have improved our work and for the support of the scope of our work.

Reviewer 2 Report

I appreciated the authors have made improvements to the manuscript, especially by adding the 16S rRNA characterization of the bacteria. In terms of the title, scope and conclusion, in line with the other reviewers, the authors should further improve the texts in an additional round of revision before being accepted for publishing.

1. The authors mentioned in line 300 that six different types of bacteria were found in the culture. It is important to reveal the 16S rRNA sequencing result for these strains. It seems to me that the most important novelty of this manuscript is about the identification of bacteria strains that can improve BPs biodegradation. This data is important and should be at least presented in the supplementary document. Also, since the authors used Nucleotide BLAST and RDP database for the analysis, their versions should be mentioned in the revised texts.

2. The authors have emphasized the importance of SCPs in inter-species interaction. However, it is more like a hypothesis and could not be well answered using the current data, making it a bit inappropriate to use it in the title. This manuscript is important in identifying the bacteria strains to improve BPs biodegradation and observing different types SCPs, either from the algae or the bacteria. These are the most important messages that the authors should highlight in the title and texts, which will prime the subsequent studies of the roles of these SCPs in BPs biodegradation or inter-species interaction. The authors can hypothesize that SCPs play important roles in BPs biodegradation and inter-species interaction when concluding the study, instead of putting it in the title. Figure S1 (please also notcie that there are two "BC" labels) is a piece of nice evidence demonstrating the authors' ability to isolate the SCPs. However, it can not tell whether these SCPs are doing something in terms of BPs biodegradation or inter-species interaction. The current data, after adding the 16S rRNA result, will be in a good shape for being published. Yet, the authors should revised the title, make the key messages more clear in the texts and try to avoid overstate about SCPs.

Author Response

Reviewer 2

We thank the Reviewer for valuable comments. Please find below point-to-point answers and descriptions of the changes made in the manuscript.

Q1. The authors mentioned in line 300 that six different types of bacteria were found in the culture. It is important to reveal the 16S rRNA sequencing result for these strains. It seems to me that the most important novelty of this manuscript is about the identification of bacteria strains that can improve BPs biodegradation. This data is important and should be at least presented in the supplementary document. Also, since the authors used Nucleotide BLAST and RDP database for the analysis, their versions should be mentioned in the revised texts.

A1. Please let me try to briefly explain what has been done as regards identification of bacteria. The experiments with bisphenols were performed couple of months ago. After performing the experiments we have kept the bacterial culture on an 1% MW-LB agar at our laboratory at the Faculty of Electrical Engineering. Due to the issues raised by the reviewer this culture was taken to the Dept. of Microbiology for PCR analysis where they were put on BD Columbia agar containing 5% sheep blood. Bacteria were initially found to have different morphologies, however, 16S rRNA analysis revealed that all variants belong to Thalassospira sp. It can reliably be concluded from these analyses that Thalassospira sp. prevailed in the culture considered and it was the only one that was identified with high confidence. Evidently our explanations in the previous version were confusing, and we hope that we have improved our messages in the new version.

We made changes in:

Results, Lines 183-185:

The bacteria were identified as Thalassospira sp. The members of the genus Thalassospira are Gram-negative rod-shaped bacteria from the Alphaproteobacteria.

Caption to Figure 4: Electron microscopy imaging of osmium-fixed samples of Thalassospira sp. SCP isolates (A-D) and P. tricornutum SCP isolates (E)….

Discussion, Lines 300-306: However, we needed a stable culture and these bacteria were successfully growing for more than one year in the same conditions. For their identification, two different colony morphologies prevailed in the BD Columbia agar (Supplementary material, Figure S2B).  Both were identified by 16S rRNA PCR to be members of Gram-negative Alphaproteobacteria, genus Thalassospira (the same percent identity for two species: Thalassospira xiamenensis and Thalassospira permensis) that are characteristic for oligotrophic waters of the East Mediterranean.

Materials and Methods, Line 446: Bacteria were identified by 16S rRNA PCR.

Materials and Methods, Lines 501-517: For identification, bacteria were cultured on BD Columbia agar containing 5% sheep blood (Becton Dickinson, Franklin Lakes, New Jersey, USA) (Supplementary material, Figure S2B) for five days at room temperature and 30 °C, respectively. Two prevailing colony morphotypes were analysed by sequencing of the 16S ribosomal RNA gene according to the manufacturer's instructions (Molzym GmbH, Bremen, Germany); the sequence of the 16S ribosomal RNA gene was visually corrected and edited and then compared with the public sequence database GenBank using the Nucleotide BLAST programme run via the server of the National Centre for Biotechnology Information NCBI (Nucleotide BLAST: Search nucleotide databases using a nucleotide query (nih.gov)). The sequences of individual strains and a contig were analysed. Sequences with a ≥ 99.0% match to a database sequence were considered to belong to the same species as the sequence with the highest similarity (Supplementary material, Table S3). In addition, all 16S ribosomal RNA gene sequences were compared to the RDP, Release 11 database (Ribosomal Database Project, RDP Release 11 -- Sequence Analysis Tools (msu.edu)), a highly curated database of aligned and annotated ribosomal RNA sequences and associated phylogenies. Each of the two variants was identified as Thalassospira sp.

Also we added raw data from 16S rRNA analysis and an image of colonies growing in BD Columbia agar with 5% sheep blood in the Supplementary material (Figures S2 and S4, respectively). We hope that we have described better what has been done in the Methods, Results and Supplementary material.

Minor:

Materials and Methods, Line 486: We added information on MW-LB Agar.

Q2. The authors have emphasized the importance of SCPs in inter-species interaction. However, it is more like a hypothesis and could not be well answered using the current data, making it a bit inappropriate to use it in the title.

A2. We agree with the reviewer. The motivation for the work was the role of SCPs, but we could not come to a decisive answer. However, a step forward has been achieved as we could observe numerous SCPs, in particular from P. tricornutum, what was not done before. SCPs stay as potential important players, in particular since no other mechanism underlying interspecies interaction has been outlined, and the mixed culture yielded the highest overall number of microalgal cells.

We suggest a new title: Interaction between microalgae P. tricornutum and bacteria Thalassospira sp. for removal of bisphenols from conditioned media

Q3. This manuscript is important in identifying the bacteria strains to improve BPs biodegradation and observing different types SCPs, either from the algae or the bacteria. These are the most important messages that the authors should highlight in the title and texts, which will prime the subsequent studies of the roles of these SCPs in BPs biodegradation or inter-species interaction.

A3. We have inserted these points to the Conclusions (Lines 602-614):

We have observed notable differences in removal of some of the BPs between the axenic P. tricornutum, mixed and Thalassospira sp. cultures. Finding the respective organisms and creating highly efficient microbe societies could be key for improvement of wastewater treatment. In particular it is important to identify the bacteria strains to improve BPs biodegradation. While the overall removal of BPs was found to be the greatest in the mixed culture which was connected to greater biomass, P. tricornutum proved excellent in removal of almost all BPC2. Our results indicate that some organisms may have ability to remove specific pollutant with high efficiency. We have visualized numerous SCPs in all samples, however, the yield of SCPs was too low to enable further characterization. The attempts to better understand the role of different types SCPs, either from the algae or the bacteria, in microorganism communities should be pursued. 

However, the result which surprised us but seems standing out is that small biomass of microalgae P. tricornutum removed almost all BPC2 (more than 99%) alone and also in co-culture (Figure 5). The data on the removal of particular BPs by the cultures considered are also of interest. However, although indicated or hypothesized, the move to downscale the complex interspecies interactions in the field of ecology to nano level by means of SCPs could be the most novel contribution to science in this work. We hope that our work will encourage also other researchers to investigate the role of SCPs in these fields. In Abstract, we have rearranged the consecutive order of our findings, the more important first. We have rewritten Conclusions (Lines 602-614).

Q4. The authors can hypothesize that SCPs play important roles in BPs biodegradation and inter-species interaction when concluding the study, instead of putting it in the title.

A4. We have removed SCPs from the title.

Q5. Figure S1 (please also notcie that there are two "BC" labels) is a piece of nice evidence demonstrating the authors' ability to isolate the SCPs.

A5. We have corrected the mistake and thank the reviewer for pointing to it.

Q6. However, it can not tell whether these SCPs are doing something in terms of BPs biodegradation or inter-species interaction. The current data, after adding the 16S rRNA result, will be in a good shape for being published. Yet, the authors should revised the title, make the key messages more clear in the texts and try to avoid overstate about SCPs.

A6. We have revised the title, abstract and conclusions. We have checked the text. We have changed a sentence at the end of Paragraph 1 (Lines 66-67). 

Reviewer 3 Report

The manuscript "Indicated role of small cellular particles in mediating inter-species interaction for removal of bisphenols from conditioned media" by Škufca D., Božič D. et al. is devoted to the biodegradation of bisphenols. It's a well-studied area and the presented results on the co-cultivation system (microalgae + bacteria) seems low-informative.

The reviewer has a lot of questions for the revised version of the manuscript. Does the detected particle affect the bisphenols level? What is the composition of the particles? What is the main products of bisphenols degradation and what is the mechanism of bisphenols utilisation by the cultures? Ironically, none from the experimental data clarified the picture at the molecular level. I think the revised version of the manuscript could not be published at the IJMS.

Here we can find only the routine observation, without any essential details or attempts to detect the mechanism of observed phenomena. Guess, it could be published as a short communication (after revision) in another appropriate journal.

Author Response

We thank the reviewer for his/hers considerations. We respectfully disagree in the essential point as given below.

Q1. The reviewer has a lot of questions for the revised version of the manuscript. Does the detected particle affect the bisphenols level? What is the composition of the particles? What is the main products of bisphenols degradation and what is the mechanism of bisphenols utilisation by the cultures? Ironically, none from the experimental data clarified the picture at the molecular level. I think the revised version of the manuscript could not be published at the IJMS. Here we can find only the routine observation, without any essential details or attempts to detect the mechanism of observed phenomena. Guess, it could be published as a short communication (after revision) in another appropriate journal.

A1. We are primarily interested in mechanisms, therefore we hypothesized that small cellular particles are the mediators of the interspecies interaction. To our best knowledge this hypothesis is novel. The fact that the interspecies interaction can be beneficial for growth of both species and also for BP removal was already found before (Prosenc et al., 2021). We argue that the interaction may not be on the molecular level and that cells communicate by means of nano-sized particles. As these are very small and dynamic structures the experimental methods, models and concepts must be revisited and if necessary new models and concepts should be developed which can hardly be done in a single article to arrive to the identity, composition and effects of SCPs. Instead it presents a potential seed for a new field.  To our best knowledge BP removal is not a solved problem in spite of the area being “well studied” and needs exploring new concepts. 

Round 3

Reviewer 3 Report

Wanna bet, the moonphases were different while the experiments had been performed. Based on this I postulate that the influence of the moon is the key factor affecting the bisphenols' utilization. It's the whole new realm of the microbilogy and new concepts and methods should be developed. However, right now we have not any precision instruments for the measurement of the moon phase and we could not to overcome this difficulties and make the clear conclusions based on the experimental data.

The authors provide the argumentation in the same manner. I guess it's not based on the data. If you can detect the small vesicles in the media, it's not automatically means that the vesicles affect the bisphenol level. The additional experiments needed to validate the hypothesis. As the simpliest variant, the authors could isolate the vesicles, add to the monocultures and test the changes after addition. Based on such experimental data we could  to discuss the idea about the nanovesicles' effect. Now it's just a words that masking the absence of the essential data.